# Heterogeneous impacts for malaria control from larviciding across villages and considerations for monitoring and evaluation

Ellie Sherrard-Smith[1,2]*, Ulrike Fillinger[3], Jean-Philippe B. Tia[4,5], Peter Winskill[2], Benjamin G. Koudou[4,5], Emile S. F. Tchicaya[5,6], Antoine Sanou[7,8], Fredros Okumu[9,10,11], Mercy Opiyo[12,13], Silas Majambere[14], Arran Hamlet[2], Giovanni Charles[2], Ben Lambert[2,15,16]‡, Thomas S. Churcher[2]‡

1 Vector Biology Department, Liverpool School of Tropical Medicine, Liverpool, United Kingdom, 2 Medical Research Centre for Global Infectious Disease Analysis, Department of Infectious Disease Epidemiology, Faculty of Medicine, Imperial College, London, United Kingdom, 3 International Centre of Insect Physiology and Ecology, Human Health Theme, Nairobi, Kenya, 4 Université Nangui Abrogoua, Abidjan, Côte d'Ivoire, 5 Centre Suisse de Recherches Scientifiques en Côte d'Ivoire, Abidjan, Côte d'Ivoire, 6 Université Péléforo Gon Coulibaly, Korhogo, Côte d'Ivoire, 7 Centre National de Recherche et de Formation sur le Paludisme, Ouagadougou, Burkina Faso, 8 University of Fada N'Gourma, Fada N'Gourma, Burkina Faso, 9 School of Biodiversity, One Health & Veterinary Medicine, College of Medical, Veterinary & Life Sciences, University of Glasgow, Glasgow, United Kingdom, 10 Environmental Health and Ecological Sciences Department, Ifakara Health Institute, Ifakara, Tanzania, 11 School of Life Science and Bioengineering, The Nelson Mandela African Institution of Science and Technology, Arusha, Tanzania, 12 Centro de Investigação em Saúde de Manhiça (CISM), Fundação Manhiça, Manhica, Mozambique, 13 Malaria Elimination Initiative, University of California San Francisco, San Francisco, California, United States of America, 14 Valent Biosciences LLC, Oslo, Norway, 15 Pandemic Sciences Institute, University of Oxford, Oxford, United Kingdom, 16 Department of Statistics, University of Oxford, Oxford, United Kingdom

‡ These authors are joint senior authors on this work.
ʘ These authors contributed equally to this work.
* ellie.sherrard-smith@lstmed.ac.uk

## Abstract

Malaria vector control tools currently focus on insecticide treated nets (ITNs) and indoor residual spraying in malaria-endemic locations, but additional preventative strategies are needed to address protection gaps. Larval source management (LSM) includes larvicide application to aquatic habitat and an array of alternative forms of environmental efforts. An individual-based transmission model for *falciparum* malaria is used to demonstrate the theoretical benefit of suppressing malaria adult mosquito vector densities through LSM. The model simulates results of epidemiological trials from Western Kenya (a hilly area with papyrus swamps adjacent to human settlements and moderate to high perennial malaria transmission) and Côte d'Ivoire (an area with Sudanese climate, reducing vegetation cover and high transmission) that applied larvicide alongside ITNs, and investigates whether estimated changes in adult density can be used to project changes in human malaria. In the Western Kenya setting generalised linear models estimate 82% (90% credible intervals: 64% − 92%) and 88% (79% − 94%) reductions in the proportion of adult *Anopheles*

**Data availability statement:** Two data resources are used, one from Fillinger et al 2009 (https://doi.org/10.2471/blt.08.055632), one from Tia et al 2024 (https://doi.org/10.1186/s12936-024-04953-8). Data are available in those manuscripts and a github repository includes all data for study one, for the more recent study of Tia et al 2024 (https://doi.org/10.1186/s12936-024-04953-8), data can be made available on reasonable request to abdoulaye.tall@csrs.ci as part of the ethics committee common to all institutions in Côte d'Ivoire. The data used in this analysis are collated as part of co-author TB's PhD thesis and will be published separately. All analysis is available in github repositories: for statistical analysis: https://github.com/EllieSherrardSmith/lsm; for transmission model: https://github.com/mrc-ide/malariasimulation.

**Funding:** ESS is funded by a UKRI Medical Research Council Fellowship (MR/T041986/1). UF was supported through ICIPE core funding by the Swedish International Development Cooperation Agency (Sida); the Swiss Agency for Development and Cooperation (SDC); the Australian Centre for International Agricultural Research, the Federal Democratic Republic of Ethiopia; and the Government of the Republic of Kenya. ESS, TSC, PW, AH, GC acknowledge funding from the MRC Centre for Global Infectious Disease Analysis (reference MR/X020258/1), jointly funded by the UK Medical Research Council (MRC) and the UK Foreign, Commonwealth & Development Office (FCDO), under the MRC/FCDO Concordat agreement and is also part of the EDCTP2 program supported by the European Union. GC acknowledges funding from the Wellcome Trust [reference 220900/Z/20/Z]. AS received funding from the Wellcome Trust [reference 222019/Z/20/Z]. The funders had no role in study design, data collection and analysis, decision to publish, or preparation of the manuscript.

**Competing interests:** The authors have declared that no competing interests exist.

*funestus* and *Anopheles gambiae* complex mosquitoes respectively as measured by CDC light traps. In Côte d'Ivoire, an 82% (56% – 93%) reduction of the dominant *An. gambiae* vector was estimated using standard window trap and pyrethrum spray catch. Both studies had variable village-level impacts. The transmission dynamics model predicted that these entomological impacts would result in a reduction in malaria prevalence in children of 6-months to 10-years of age of 48 – 72% in Kenya, and a 11 – 78% reduction in all-age clinical incidence across villages in Côte d'Ivoire, which are broadly consistent with the empirically observed outcomes. High heterogeneity between villages within the same study indicate that the relative or absolute reductions in mosquito adult density observed in these trials cannot be simply extrapolated to other regions. The LSM strategy adopted, unit area covered, and multiple environmental covariates all contribute to differences in indicators that could be used to assess entomological impacts and the corresponding epidemiological outcomes. This important malaria control tool was impactful across all sites examined, though further work is needed to understand how best to use this tool in the fight against malaria.

## Author summary

Additional preventative strategies are needed to address protection gaps in malaria control. One potential opportunity is larval source management (LSM), which reduces the number of mosquito adults emerging by modifying, treating or removing aquatic breeding sites. Measuring the impact from LSM can be challenging because the LSM strategy adopted, the effort deployed (for example, unit area covered), and the environment (how many breeding sites there are) can all contribute to how LSM can change the number of mosquitoes in an area and the clinical malaria cases averted. We estimated the impact from the application of larvicide conducted in two contrasting ecological settings in Kenya and Côte d'Ivoire by fitting difference-in-differences statistical models to adult vector densities before and after the application started. Differences in adult mosquito densities between the control and intervention villages were sufficient to parameterise a *Plasmodium falciparum* malaria parasite transmission dynamics model and broadly reproduce epidemiological trial outcomes. This work highlighted the variability in potential impact from LSM and provides considerations for surveillance of critical indicators for LSM performance monitoring to facilitate impact interpretation through modelling.

## Introduction

Across Africa – where more than 90% of malaria-associated deaths occur annually – insecticide treated nets (ITNs), and the prompt treatment of clinical cases are key

interventions used to control transmission [1], while new vaccines are in the process of being scaled [2,3]. It is widely recognized that these interventions alone will be unable to push malaria to elimination on the African continent [4–6]. But impacts will differ spatially as malaria transmission is dependent on the local behavior of communities, variable immunity and disease endemicity, and local mosquito ecology [7,8]. The integration of other interventions is needed to achieve elimination.

Generally, ITNs are delivered by national campaigns roughly every 3 years and work best immediately after installation. Effects then wane as the insecticide potency reduces and holes accrue, and nets get lost or discarded over time so fewer people use them later in the campaign cycle [9]. Average ITN use remains around 40–50%, far lower than the high access achieved immediately after deployment, though trends indicate a slight increase in people sleeping under nets in the last year, 2023 [7]. Efforts are made to supplement mass campaigns through school-based or clinic-based top up opportunities [10–12]. There is some evidence of an increase in malaria parasite transmission occurring in the hours of a day when indoor interventions cannot prevent bites and kill vectors [13–16]; this contributes to residual transmission. Together these circumstances leave protection gaps that grow over the 3-year cycle of ITN deployment.

Larval source management (LSM) is an umbrella term for using larvicides, modifying or manipulating the habitat, or eliminating or removing habitats that are suitable for mosquito larvae [17]. Historically, source reduction through larval control has been empirically shown to suppress malaria transmission in observational studies [18–21] but rare exceptions exist, for example, in the flood plain of the Gambian river where breeding sites are rife and proved inaccessible by ground teams during a larviciding trial in 2006–2007 [22]. Reviews [18,23,24] evaluate pre-1940s large-scale environmental modification that successfully eliminated malaria vectors, for example, from Italy [25], the Tennessee valley, USA [26], Malaysia [27] and around copper mines in Zambia [28]. The arrival of DDT in the 1940s resulted in many programs transitioning from larval or habitat management strategies to spraying as the primary control method for anopheline vectors. During the first attempt at malaria eradication, which ended in 1969, many methods were developed for malaria control which centered around the use of insecticides. Supply chains were poor to deliver interventions to communities who were also battling politically to regain their independence from Europeans [29]. The challenge was probably exacerbated by ecological changes in land use that benefited *Anopheles* mosquitoes [30,31]. Long-lasting ITNs are attributed with successful control in many areas from 2000 onward [1], but challenges to access and maximise the use of ITNs across communities, and resistant vectors mean that this core intervention may be reaching a limit on what it can achieve.

Community-driven management of larval habitat, together with scale-up of vector interventions and educational programs encouraging ITN use, was associated with a reduction in infection prevalence from 24% to 10% in children in Malindi, Kenya [32]. Intensive community engagement efforts have seen similar benefits in Malawi [33]. However, integrated vector management has not yet been widely adopted to sustainably improve malaria control, particularly across countries within malaria-endemic Africa. This may, in part, be because, to-date, very few trials have been conducted [34–36]. The WHO guidelines report that no recommendation is possible for habitat manipulation or modification, nor for larvivorous fish because empirical data are insufficient or absent [17]. The authors are unaware of any cluster-randomized control trials with epidemiological endpoints – recognized as the gold standard – that have been conducted in Africa on any form of LSM [17,36]. In addition, it is not clear how best to measure impact [37]. The most widely used metric considers reductions in mosquito densities of either larvae, pupae or adults. The epidemiological impact from this change will depend on the location in part because LSM offers a variety of more-or-less permanent strategies.

For any attempt to control malaria transmission, it is highly likely that conditions vary between different sites: different seasonal patterns in vector behaviours will exist – breeding sites can be ephemeral and not fixed in number or location, sporozoite rates and infectious reservoirs will differ, and house structures, use of ITNs or other commodities, access to care and outdoor activities will all contribute to shaping the challenge. Most vector control programmes have a conceptually simple method of assessing the use of an intervention. For example, IRS is assessed as the percentage of eligible structures that are sprayed whilst ITN use and access can be established from population surveys. LSM is more complex

as its success depends on the percentage of larval sites treated which, traditionally, has required reliable estimates of the number of larval sites in an area which is unknown and hard to estimate unless such sites are rare. New technologies may resolve, or reduce, this challenge by enabling the delivery of larvicide to an area without all sites being identified [38,39]. Even so, for a given amount of effort, the achievable outcome will likely differ between sites; for example, if it is possible to treat 100 larval sites in a day, the relative reduction for a community with 125 breeding sites would be 80%, and 10-fold better than a corresponding location with 1,250 equivalently productive sites. Differences will likely exist in the productivity of different habitats and their findability, and teams will be built given the environmental conditions. It is also hard to sufficiently power trials for adult mosquito density abundance, so this example is an oversimplification but serves as a start point to explore potential impacts. Measuring the absolute and relative reduction in larval sites treated will likely have different epidemiological impacts in different settings. Similarly, absolute and relative reductions in mosquito abundance are regularly used to quantify the entomological efficacy of LSM. It is unclear whether these measures could both inform on the potential efficacy and corresponding cost-effectiveness of LSM and be more informative than estimates of the percentage of larval sites treated.

In this contribution, we use a transmission model to show the theoretical benefits of LSM in a range of different epidemiological and entomological settings. The model simulates the suppression of adult female mosquito densities and tracks the corresponding changes in malaria burden in humans. It is then used to explore whether empirical estimates of mosquito adult or larval abundance can reliably predict changes in malaria burden in two different village-level trials. The two trials took place in Western Kenya between 2004 and 2006 [34] and Côte d'Ivoire between 2019–2020 [35]. The site in Western Kenya worked within 6 highland valley communities (1453 – 1632 metres above sea-level) in a region that had undergone recent clearance of natural swamps and deforestation for crop cultivation leaving exposed water bodies ideal for mosquito habitat [34]. Villages were built within the valleys with small streams and papyrus swamps being common features [34]. Transmission was perennial with vector densities generally tracking the seasonal pattern of long rains from March to June and shorter rains October to December [34]. The sites in Côte d'Ivoire received 1200 – 1400 mm of rainfall annually, with temperatures from 21 – 35°C, and followed a 6-month dry (November to April) and rainy (May to October) season [35]. The vegetation cover is being lost in the region as a consequence of climate change and extended dry seasons [35]. Adult mosquito densities were tracked in the trials to assess the association between mosquito reductions and epidemiological impacts on malaria prevalence [34] or annual incidence [35] during the application of *Bacillus thuringiensis israelensis* (*Bti*)-larviciding. We also had larval data for the Cote d'Ivoire trial but were unable to access this for the Kenyan trial. Both studies compared the use of larviciding in areas with ongoing ITN use, but trials deployed the larvicide at different times across an approximate 3-yearly mass ITN campaign cycle. In Kenya, ITNs were deployed in an area with no historical use of ITNs as larviciding began, with small but continual increases in the numbers of people using nets recorded through the trial. In Côte d'Ivoire, larviciding took place about 2 years after the most recent mass ITN campaign. Epidemiological outcomes were tracked differently in the two trials. In Kenya, the trial team carried out cross-sectional prevalence surveys every two months throughout the baseline and follow-up. In Cote d'Ivoire, health facility data were aggregated retrospectively across the baseline and follow up year to investigate any observable changes in burden.

## Results

### LSM potential

First, we theoretically explored how the outcomes from LSM efforts might impact different ecological settings. In these hypothetical scenarios, local mosquitoes are assumed to be fully susceptible to pyrethroid insecticide meaning that the impacts from ITNs would be optimal according to estimates of their efficacy [40]. Simulations are parameterized so that 87% of mosquito foraging attempts for blood meals take place during bed time hours. The impact from ITNs used by 60% of the population immediately after deployment, with waning impact over time, is compared to 60% ITN use plus

larviciding efforts that either: i) suppress mosquitoes densities by 60% as LSM starts, or; ii) reduce the absolute number of mosquitoes by the same number as a 60% reduction in the low transmission setting. This results in lower relative reductions in the moderate and high transmission settings investigated. (The value of the percentage reduction achieved in the low transmission setting is arbitrarily selected for illustrative purposes but allows us to estimate absolute vectors per person from the model.) The model projects that LSM which successfully reduces mosquito density can cause a substantial reduction in malaria burden. It is likely that there is a ceiling on the amount of breeding sites that can be treated per unit time and workforce depending on LSM strategy. The resulting impact shown is conditional upon model assumptions and shows how the impact of LSM could vary substantially between sites. LSM is assumed to achieve a defined reduction in adult mosquito emergence, and efforts maintain this reduction over time once initiated, so the outputs could represent different LSM (larvicides, modifying/manipulating the habitat or eliminating/removing the habitats). In these theoretical simulations, only mosquitoes that have behaviours reflecting an *Anopheles gambiae*-like species are simulated (the biting rate is assumed to be 0.33 per day, the background mortality rate to be 0.132 per day, and 92% of mosquitoes to seek a blood meal on a human host).

In the low transmission setting where a single seasonal peak in transmission is simulated (Fig 1A), the average number of female mosquitoes per person across 3-years prior to the intervention was 3.12 vectors per person (this is the ratio of female mosquitoes to people tracked in the simulation), and across the 3-years afterward this number was 1.26 vectors per person, indicating the number of mosquitoes reduced by 1.86 vectors per person. This reduction equates to a 60% relative reduction in adult mosquito densities (Fig 1B) and is predicted to avert 93% of all-age clinical cases in the low transmission scenario (Fig 1B). A 60% relative reduction in female mosquitoes achieved through LSM on top of ITNs in the low transmission setting is predicted to reduce malaria prevalence in children under 5-years of age by 67% – 85% relative to no interventions across a range of estimates for the proportion of vector bites assumed to be received during the time people were in bed ranged from 40% – 99% (Fig 1C). The removal of the same absolute number of mosquitoes per person from a moderate (Fig 1D–1F) and high (Fig 1G–1I) transmission setting would correspond to a smaller relative reduction in mosquito densities of 12.4% and 3.4% respectively. These results depend on an assumption that breeding sites are equivalently productive, i.e., efforts to treat sites reduce the numbers of adult vectors by an absolute number, whereas in reality, highly productive sites may be more easily found and thus these estimates may correspond to a lower bound on potential impact. For the combined ITN and LSM intervention, this is predicted to only reduce prevalence in the same age group by 23% – 42% and 9% – 32% across the residual transmission range explored (40% - 99% of bites taken when a person is in bed, Fig 1F and 1I respectively) relative to no intervention. This translates to 61% and 40% of all-age clinical cases averted in the moderate (Fig 1E) and high (Fig 1H) transmission scenarios respectively. Conversely, if LSM in addition to ITN campaigns in moderate and high transmission settings reduced the relative number of mosquitoes by 60%, the transmission model predicted the removal of 7.3 and 24.7 mosquitoes per person, respectively (note these numbers are estimated from the transmission model and are used to demonstrate the difference for absolute and relative impacts). The models predict this 60% relative reduction could cause a 47% – 71% reduction in prevalence, or 82% relative reduction in all-age clinical cases, in the moderate transmission area, and a 31% – 54% reduction in prevalence, or 49% relative reduction in all-age clinical cases, in the high transmission setting. A sensitivity analysis for different resistance profiles – that, in the transmission model, infer lower efficacy of ITNs – is shown in Fig A in S1 Text.

The relative impact of LSM when conducted in an area with ITNs was greatest when indoor mosquito biting was lowest (Fig 1C, 1F and 1I). In these simulations ITN usage is assumed to always reach 60% after deployment and wanes thereafter. In our comparisons (that alter the proportion of bites assumed to be taken during bed time), ITNs provide increased protection to the community as proportionally more mosquitoes forage for blood meals indoors when people are using the ITNs, so LSM can have greater benefit by reducing mosquito densities and bites when ITNs are not able to kill as many vectors. Similarly, the epidemiological efficacy of ITNs is reduced as the level of pyrethroid resistance in the local mosquito population increases (Fig A in S1 Text) [41–43]. Our assumptions are that resistance in vector populations would not

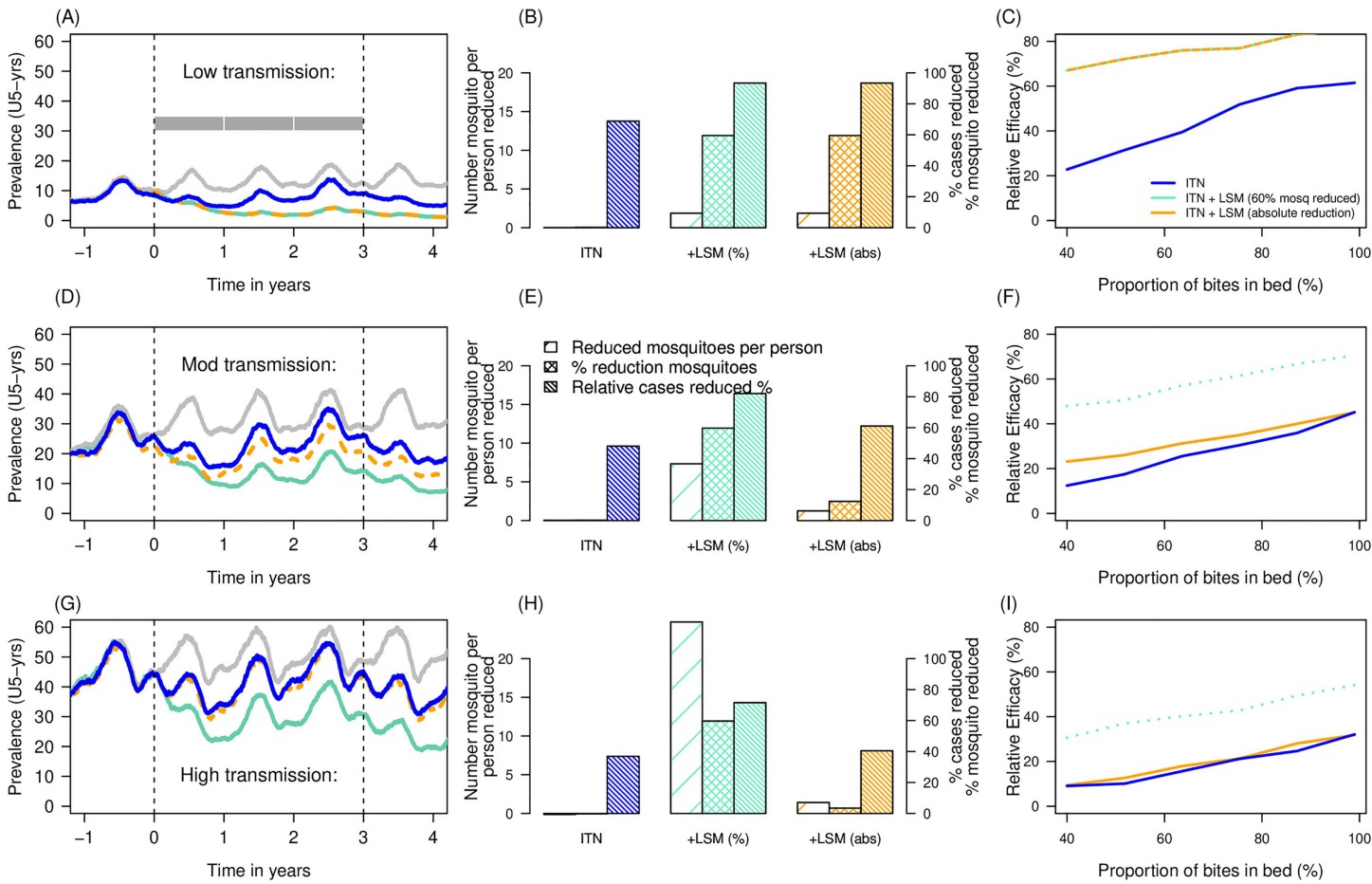

**Fig 1. Theoretical impact of larval source management strategies that suppress *Anopheles* mosquitoes.** Panels A, D and G show the prevalence in children of 6 to 59 months of age as measured by microscopy over time when endemic malaria burden is low, moderate or high respectively. In these scenarios, local mosquitoes are assumed to be fully susceptible to pyrethroid insecticide. Simulations are parameterized so that 87.2% of mosquito foraging attempts for blood meals take place in bed. The impact from ITNs used by 60% of the population with waning impact over time (blue line) is compared to 60% ITN use plus larviciding efforts that either: i) suppress mosquitoes densities by 60% (green line) from time 0 onward, or; ii) reduce the absolute number of mosquitoes by the same number as a 60% reduction in the low transmission setting (orange line). Outcomes are compared to a counterfactual simulation (grey line) were no new interventions are deployed from time 0. Panels B, E and H: Bar charts reporting the simulated changes in mosquito density and resulting epidemiological impact of LSM over 3 years. Bar colours are the same as panels A, D and G, with blue bars show impact of ITNs only, with the green and yellow bars respectively indicating the relative (percentage change, right axis), and absolute (in number of mosquitoes per person, left axis) impact of deploying LSM. As before, the model is parameterized to simulate a scenario of deploying ITNs and LSM where 60% reductions in vector densities are simulated (green), or the removal of the absolute number of mosquitoes (orange) that aligns with the low transmission scenario (thus, green and orange bars match in panel 1B). Fine hatching shows model predictions for the relative reduction in malaria cases (right axis), be it for ITNs only (blue), or the relative (green) or absolute (yellow) impact of LSM. Panels C, F and I: Relative efficacy over 3 years [relative reduction in parasite prevalence in children under 5 years of age compared to no intervention] when deploying ITNs (blue), ITNs with LSM that reduces 60% of mosquitoes (green), or ITNs with LSM reducing the same number of mosquitoes as would achieve a 60% reduction in the low transmission scenario (orange) versus different levels of residual transmission (determined by the propensity of mosquitoes to feed indoors).

change the impact of LSM on mosquito emergence and therefore we see a constant benefit from LSM regardless of the level of pyrethroid resistance (Fig A in S1 Text).

In all the transmission settings, the relative additional benefit from larviciding (the area between the blue (ITNs only) and green (relative reduction) or orange (absolute reduction) lines each year, Fig 1) is greatest in the third year of the mass ITN campaign when the nets are performing least optimally (Fig 1A, 1D and 1G) but this is because the simulations

assume the effect from the LSM action is sustained, while the entomological efficacy and population use of ITNs wane. If LSM efforts were removed, we would expect resurgence in mosquito densities and consequently malaria cases, as for any intervention.

## Simulating outcomes from epidemiological trials using larviciding

**Kenya.** Data in the Kenyan trial came from 6 sentinel villages (Fig B in S1 Text) and reductions in adult mosquito densities were observed across all the sites (larviciding, 84 – 99% relative reductions in adult mosquito density, and non-larviciding, 80 – 93% relative reductions comparing before and after trial interventions were implemented, Table 1) [34]. Standard pyrethroid-treated nets (ITNs, PermaNet 2.0 and Olyset Nets) were gradually introduced in all communities, in addition to the intervention in the trial. The larviciding and non-larviciding arms of the trial were selected so that the baseline number of aquatic habitats were roughly equal in communities in each arm. Larval and adult mosquito densities were monitored during the year prior to larviciding and throughout the trial. The larviciding communities received weekly larviciding treatments to all accessible habitats. Statistically, using a difference-in-differences analysis, an estimated reduction of 82% (64% – 92%) in adult *An. funestus* densities and 88% (79% – 94%) in adult *An. gambiae* densities, was associated with larviciding at trial arm-level (Table 1). The village level differences showed that in Musilongo, *An. funestus* was reduced by 87% (25% – 99%) while *An. gambiae* densities reduced by 97% (92% – 99%). In Kezege, these

**Table 1. The empirical estimates and derived parameters used to calibrate the transmission model for villages testing nets or nets and larviciding in the Kenyan trial [34]. Crude estimates for the total reduction in adult mosquitoes at the village level, the global estimates of species reductions according to the GLMM analysis (Eqs 4–7). The relative species compositions in each village that are used to weight the percentage reductions in adult densities simulated in the transmission model, the village-level statistically estimated ranges in reductions in vectors, the baseline prevalences as measured using microscopy in the baseline cross-sectional surveys of the trial, and information on net use and how this changed throughout the trial (Fig C in S1 Text). Full parameters are provided in S1 Data.**

| | | Sentinel villages simulated by transmission model | | | | | |
|---|---|---|---|---|---|---|---|
| **Village names** | | **Musilongo** | **Kimingini** | **Kezege** | **Wamondo** | **Emutete** | **Wakikuyu** |
| **Parameter** | | **1 (*Bti*)** | **2** | **3 (*Bti*)** | **4 (*Bti*)** | **5** | **6** |
| Total reduction in *An. gambiae* s.l | | 98.9% | 81.6% | 95.3% | 84.1% | 81.4% | 82.5% |
| Total reduction in *An. funestus* s.l | | 94.6% | 92.5% | 94.5% | 96.6% | 88.7% | 80.1% |
| Model estimated reduction explicitly due to LSM: $\partial_{DiD,funestus}$ $\partial_{DiD,gambiae}$ | | 82.3% (63.6% − 91.6%) *88.1% (78.6% - 93.9%)* | | | | | |
| Proportion *An. gambiae* s.l. | | 73.4% | 79.7% | 59.5% | 38.7% | 66.3% | 76.4% |
| Proportion *An. funestus* s.l. | | 26.6% | 20.3% | 40.5% | 61.3% | 33.7% | 23.6% |
| *Impact explicit to larviciding (90%CrI) | *An. gambiae*: *An. funestus*: | 0.97 (0.92 − 0.99) 0.87 (0.25 − 0.99) | 0 0 | 0.78 (0.46 − 0.92) 0.62 (-0.04 − 0.88) | 47.2 (-0.42 − 0.82) 0.13 (0.07 − 0.21) | 0 0 | 0 0 |
| Baseline prevalence in children of 6-months to 10-years of age (measured using microscopy): range: Feb 2004 - Jun 2005 | | 63.3% (50.7 -70.0) | 60.1% (42.7 − 72.7) | 55.8% (41.7–77.1) | 25.7% (12.6 − 41.4) | 44.9% (15.4 − 66.7) | 40.2% (18.9 − 63.3) |
| Net use increased through the study (Fig C in S1 Text), here, average annual net use reported for each trial arm is shown | | | | | | | |
| Historic net use | | 12.3% | 8.1% | 20.4% | 21.7% | 15.8% | 9.4% |
| July 2005–2006 | | 37.8% | 15.8% | 30.4% | 35.4% | 25.7% | 27.8% |
| July 2006 – July 2007 | | 54.3% | 36.9% | 49.8% | 48.4% | 43.5% | 43.5% |

Full parameter inputs are provided in S1 Data.

reductions were estimated to be 63% (-4% – 88%) and 78% (46% – 92%) for *An. funestus* and *An. gambiae* respectively. In Wamondo, the reductions were estimated to be 13% (7% – 21%) and 47% (-42% – 82%) for *An. funestus* and *An. gambiae* respectively (Fig 2). Statistical model diagnostics are shown in Table A in S1 Text.

The modelled slide-positive prevalence in children of 6-months to 10-years of age is broadly consistent with the observed data across the 6 villages in the Kenyan trial (Fig 3). The transmission model is calibrated to the mean estimated prevalence at baseline across the previous year (grey polygons preceding onset of larviciding in Fig 3A–3C and Fig 3E–3G). After the trial begins, the simulation reflects the model estimated trend in prevalence given changing ITN use and larvicide impact in each of the respective villages (Table 1 for parameter estimates). The reported prevalence in each sentinel village that was measured during the trial are overlaid. Fig 3D shows that the direct comparison of reported prevalence and transmission model simulations for matched time points and age cohorts with estimates most accurately recreating the situation in Musilongo (Fig 3A). In each village, a counterfactual simulation is shown that reflects the model estimated trend were larviciding not deployed. The reported prevalence in each village was consistently lower than the model estimates in both the larviciding and control villages. Fig 3G shows the absolute difference between the model estimates and the reported prevalence for each village. Using these calculations, there was an estimated 12% (90%

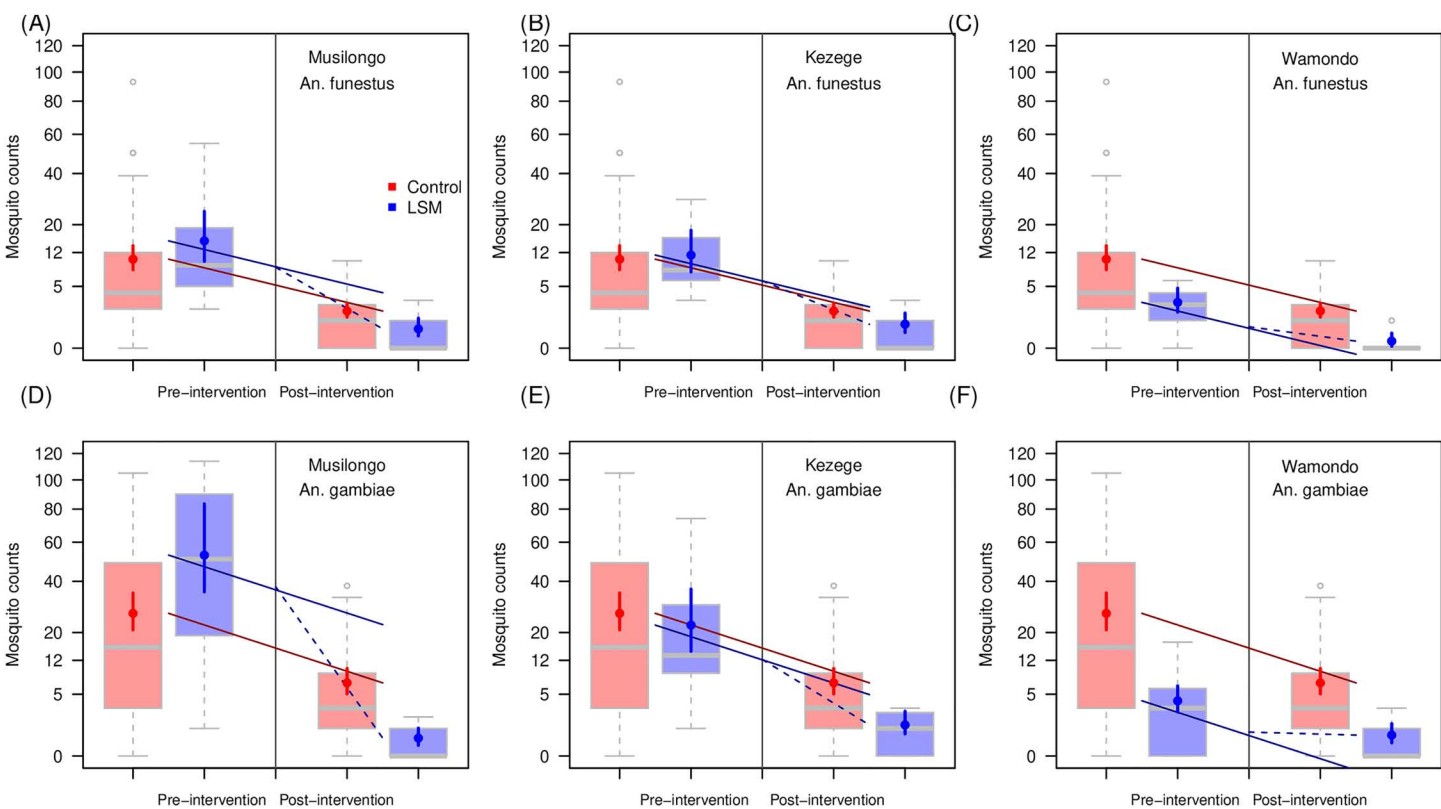

**Fig 2. Village-level changes in mosquito counts before and after intervention for Kenya at the village level [34].** In each panel, boxplots show the empirical data mosquito count ranges for, in order, the combined control sites prior to July 2005 (treatment initiation date), the corresponding larviciding village (Musilongo, panels A and D; Kezege, panels B and E; and Wamondo, panels C and F), the post-intervention control data, and the corresponding larviciding site data post-intervention. The statistical estimates from the difference-in-differences analysis (vertical lines show 90% credible intervals with point estimate for median) are overlain. The top row shows changes in *An. funestus* mosquito counts at the village level, the lower row shows those for *An. gambiae*. The straight red lines indicate the change from pre to post intervention in the control villages, the solid blue line indicates the same amount of change adjusted for the pre-intervention mosquito counts. The dashed blue line indicates the additional benefit from the LSM.

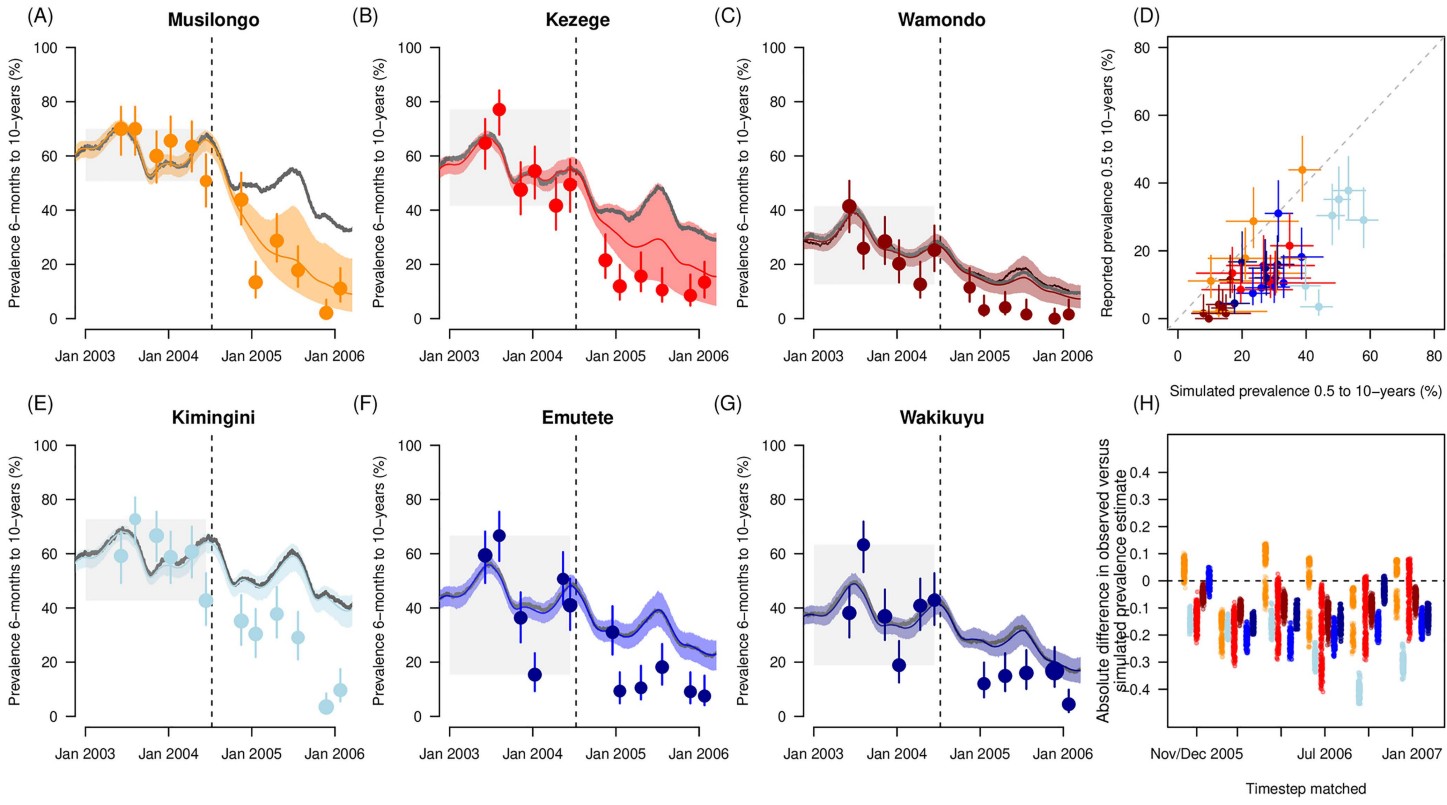

**Fig 3. Model simulations for each sentinel village, Kenya.** A-C & E-G) Prevalence trends in children of 6-months to 10-years of age are simulated in the transmission model using parameters to match site specific conditions, net use as reported following the mass net distribution and top up interventions (Fig C in S1 Text), and onset of larviciding in recipient villages (indicated by vertical dashed line in A-C, E-G), as noted in Table 1. Grey lines show counterfactuals where no reduction in mosquito densities is simulated. This corresponds to a scenario with ITNs only and mean estimates for model parameters. Polygons show the range in outcomes given the sensitivity analysis across model parameters that considered human biting preference, propensity to bite indoors and uncertainty in ITN and larviciding impact. Trial reported prevalence estimates are shown as points, their size indicates the sample size for each cohort tested at cross-sectional survey, with 95% binomial confidence intervals. Model is calibrated to the average prevalence observed during baseline year (grey shaded polygons). **(D)** The slide-positive parasite prevalence in children of 6-months to 10-years of age matched for each cross-sectional survey timestep. Points lying on the one-to-one line indicate perfect prediction. Ranges in predictions are shown for the transmission model estimates and binomial confidence intervals on empirical data. **(H)** Comparison of the trial and model simulated efficacy at each cross-sectional survey relative to the mean prevalence across the baseline period. In all panels, colours denote different villages (larviciding villages: Musilongo (orange), Kezege (red) and Wamondo (dark red); ITN only villages: Kimingini (light blue), Emutete (blue) and Wakiyuku (dark blue)).

uncertainty interval: 3 – 18%) underestimate in the model prevalence outcomes compared to the surveyed data observed in both control and treatment villages.

There were considerable differences in the scale of the relative and absolute impact between villages within the same trial. The relative reductions in *An. gambiae* densities of 97% (90%CrI: 92% – 99%), 78% (90%CrI: 46% – 92%) and 47% (90%CrI: -42% – 82%), in Musilongo, Kezege and Wamondo corresponded to absolute reductions in female mosquito numbers per person in the transmission model of 41.8 (90%CrI: 39.3 – 43.7), 17.2 (90%CrI: 11.0 – 20.1), and 1.2 (90%CrI: -0.8 – 1.9) vectors per person in the respective villages. The corresponding relative reductions for female *An. funestus* of 87% (25% – 99%), 62% (-0.04% – 88%) and 13% (7% – 21%) corresponded to 14 (90%CrI: 6 – 16), 10 (90%CrI: 0 – 13) and 0.9 (90%CrI: 0.6 – 1.1) absolute reductions in the mean number of female vectors per person per year in Musilongo, Kezege and Wamondo respectively.

Using the transmission model results, the mean relative efficacy against prevalence comparing the simulated result given the added benefit of larvicide to ITNs relative to the counterfactual (no larviciding, ITNs alone) for Musilongo, Kezege and Wamondo, was 67.8% (62.7% – 71.9%), 57.1% (48.6% – 65.5%) and 59.5% (48.4% – 70.8%) respectively.

**Côte d'Ivoire.** Data from Côte d'Ivoire came from 4 villages (2 were treated with LSM and 2 were monitored as comparisons) in the department of Napiélédougou in Korhogo Health District where communities had received pyrethroid-only ITNs (PermaNet 2.0) via the mass campaign that took place in 2017 [35]. Larviciding was conducted fortnightly outside the times of heavy rain with larviciding treatment events occurring across the follow-up year [35]. The measured relative reduction in adult mosquito densities attributed to larviciding according to the difference-in-difference analysis was 82.0% (55.9% – 92.6%) at the arm level (Table 2). In Kakologo, the estimated reduction was 90.0% (69.8% – 96.8%), while in Nambatiourkaha it was 49.9% (-60.7% – 85.3%). Fitting the difference-in-differences statistical model to the larval data returned a relative reduction of 98.5% (95.0 – 99.6%) in the larviciding settings compared to the ITNs alone. In

**Table 2. The data estimates and derived parameters used to calibrate the transmission model for an ITN-only and ITN plus larviciding trial in Côte d'Ivoire [35]. Crude estimates for the total reduction in larval and adult mosquitoes at the village level (derived from empirical data), the global estimates of vector density reductions according to the GLMM analysis (Eqs 4–7). The relative species compositions in each village that are used to weight the percentage reductions in adult densities simulated in the transmission model, the village-level statistically esti-mated ranges in reductions in vectors, the baseline and follow up annual all-age incidence measured across the baseline year using health facility data, corresponding case counts for village level data, and information on net use and pyrethroid resistance of local vectors.**

| | | Sentinel villages simulated by transmission model | | | |
|---|---|---|---|---|---|
| Village names | | Kolékaha | Lofinékaha | Kakologo | Nambatiourkaha |
| Parameter | Description | 1 | 2 | 3 (*Bti*) | 4 (*Bti*) |
| Baseline counts: *Anopheles* larvae (early and late) | | 501 | 0 | 672 | 153 |
| Baseline counts: *Anopheles* adults | | 47 *An. gambiae* | 4 *An. gambiae* | 36 *An. gambiae* 1 *An. zeimanni* | 11 *An. gambiae* |
| Model estimated reduction explicitly due to LSM: $\partial_{DiD}$ | | Overall (larval stages): 0.98 (0.95 – 0.99) Overall (adult vectors): 0.82 (0.56 – 0.93) | | | |
| Proportion species | | | | | |
| An. gambiae s.l. | | 0.980 | 0.953 | 0.880 | 0.835 |
| An. funestus s.l. | | 0.000 | 0.040 | 0.000 | 0.139 |
| other species | | 0.020 | 0.007 | 0.120 | 0.026 |
| *Impact of larviciding $\partial_{DiD}$ (90%CrI) | Adults: Larvae: | 0 0 | 0 0 | 0.90 (0.70 – 0.97) 0.99 (0.96 – 1.00) | 0.50 (-0.60 – 0.85) 0.92 (0.71 – 0.98) |
| Baseline incidence estimated (cases per 1,000 people): | | Under 5 years: 88.1 5 – 15 years: 55.3 Over 15 years: 44.6 | | Under 5 years: 136.9 5 – 15 years: 87.3 Over 15 years: 67.5 | |
| All-age malaria incidence for baseline year (cases per 1,000 per year): March 2018 to February 2020 | | 188.0 | | 291.8 | |
| All-age malaria incidence in follow up year (cases per 1,000 per year): March 2019 to February 2020 | | 186.1 | | 111.4 | |
| Baseline cases/ number of people | | 278/ 496 | 215/ 553 | 426/ 620 | 339/ 766 |
| Total cases during follow up | | 321 | 167 | 186 | 106 |
| ITN use (over 1 year follow up) | | 36.2% | | 40.7% | |
| Pyrethroid resistance (for Pyrethroid-ITN efficacy) | | 60 – 95% | | 60 – 95% | |
| Treatment initiation | | None conducted (counterfactual) | | March 2019 – February 2020 | |

Kakologo, the reduction in larval *Anopheles* was 98.9% (96.1% – 99.7%) and in Nambatiourkaha the equivalent reduction was 92.6% (69.0% – 98.4%) (Fig 4). Model diagnostics are shown in Table B in S1 Text and trial arm estimates shown in Fig D in S1 Text.

The transmission model simulations explored a relative reduction of adult female mosquitoes across these ranges for the two treatment villages. The burden – estimated from Health Facility data – in the treatment arm was substantially higher at baseline where health facility all-age incidence estimates were reportedly 291.8 cases per 1,000 population annually (March 2018 – February 2019) and this compared to 188.0 cases per 1,000 population annually in the ITN only arm [35]. Using the scale of difference, to compare with the Kenyan work, we translated the village level counts of cases in the baseline year given population sizes to estimate all-age prevalence trends at the village level (Methods). This indicated a potential reduction in all-age prevalence of 34.5% (16.1% – 51.6%) in Kakologo (Fig 5A) and 16.1% (-21% – 51%) in Nambatiourkaha respectively (Fig 5B).

The model estimated all-age absolute and relative cases averted during follow-up year are compared to the simulation baseline year to align with the trial publication [35]. An estimated 662 (90% uncertainty interval: 349 – 838 cases per 1,000 people) fewer cases per 1,000 people (a relative reduction of 78%, 90%CrI: 41% – 99%) was estimated from the model outcomes in Kakologo. In Nambatiourkaha, there were an estimated 60 (90%CrI: -694 – 511) fewer cases per 1,000 people (relative reduction of 11%, but uncertainty was high 90%CrI: -127% – 94%) calculated using the model. The trial empirical work recorded 180 fewer cases per 1,000 people from baseline to follow up year for the larvicide arm of the trial – a 62% relative reduction, which falls between the village level estimates from the model outputs.

Similar to in Kenya there is considerable heterogeneous impacts between the 4 villages. The relative reductions in the adult densities of the dominant species *An. gambiae* of 90% (90%CrI: 70% – 97%) and 50% (90%CrI: -60% – 85%) in Kakologo and Nambatiourkaha corresponded to absolute reductions in female mosquito numbers per person according to the transmission model of 47.2 (90%CrI: 33.2 – 51.9) and 3.9 (90%CrI: -5.3 – 6.8) vectors per person respectively. This exercise was repeated for larval density estimates (Fig E in S1 Text).

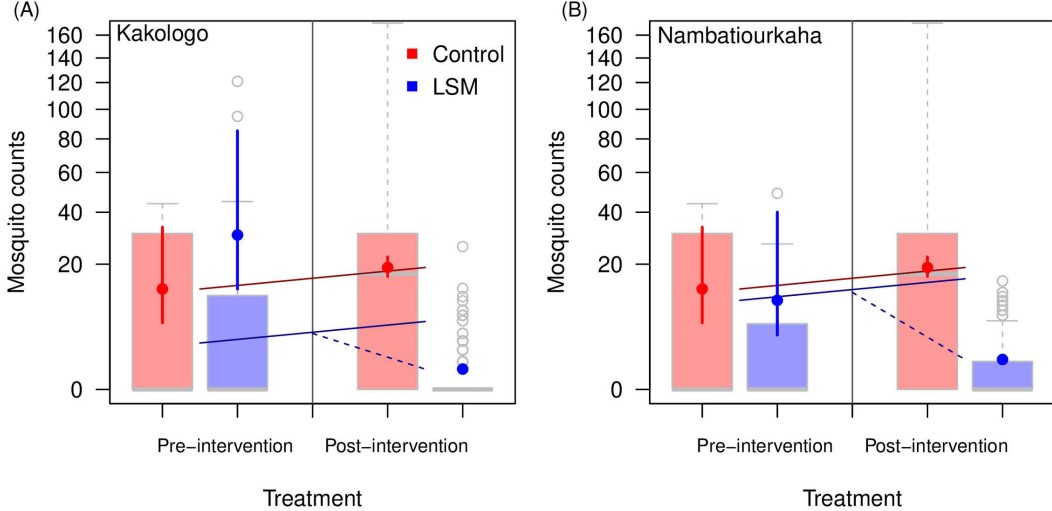

**Fig 4. Village-level changes in mosquito counts before and after intervention for Côte d'Ivoire at the village level [35].** In each panel, boxplots show the empirical data mosquito count ranges for, in order, the combined control sites prior to treatment initiation, the corresponding larviciding village (Kakologo, panels A; Nambatiourkaha, panels **B)**, the post-intervention control data, and the corresponding larviciding site data post-intervention. The statistical estimates from the difference-in-differences analysis (vertical lines show 90% credible intervals with point estimate for median) are overlain. The straight red lines indicate the change from pre to post intervention in the control villages, the solid blue line indicates the same amount of change adjusted for the pre-intervention mosquito counts. The dashed blue line indicates the additional benefit from the LSM.

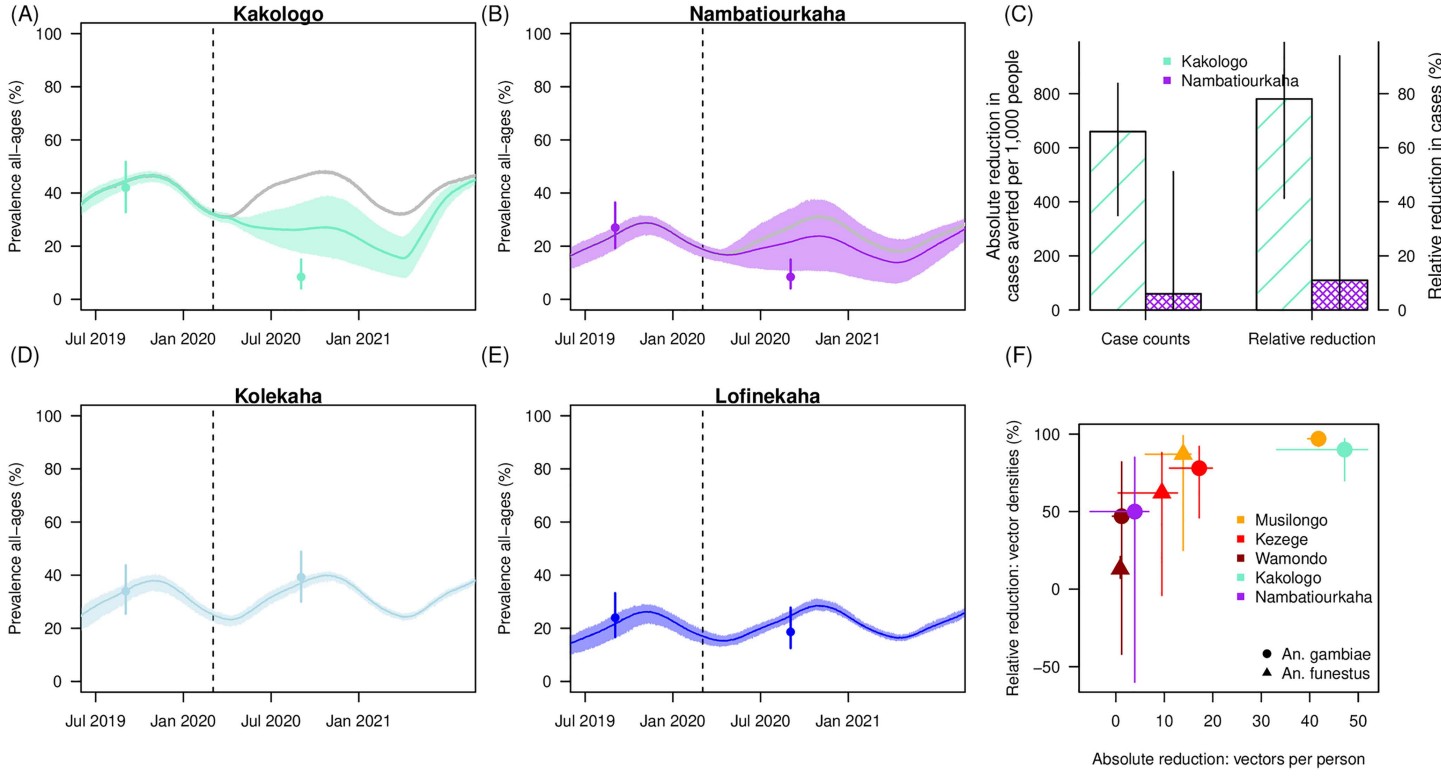

**Fig 5. The model simulated results from recreating the Côte d'Ivoire trial.** (A-B and D-E) Model simulated estimates of all-age prevalence in the villages of Côte d'Ivoire. In treatment villages (A: Kakologo, and B: Nambatiourkaha), grey lines indicate the counterfactual where parameters are matched but no effect from larvicide is simulated. Estimated prevalence from trial information are shown but were never empirically measured, shown here to serve as an indicator of the capacity of the model to broadly reproduce the scale of impact observed using the health facility data from the trial [35]. Uncertainty intervals from model parameter sensitivity analysis shown in A-B and D-E as polygons around mean estimated prevalence trend. **C)** Absolute (left axis), and relative (right axis) estimates of the reduction in clinical cases in the follow-up year in comparison to the baseline year for each treatment village Kakologo (light green), and Nambatiourkaha (purple). **F)** Association between relative reduction in adult mosquito densities and absolute reduction in mosquito numbers per person per year as estimated by the transmission model outputs. Colours show each treatment village: in Kenya, Musilongo (orange), Kezege (red) and Wamondo (dark red); and in Côte d'Ivoire, Kakologo (light green) and Nambatiourkaha (purple). Symbols indicate mosquito species. 90% uncertainty interval shown in C and **F.**

Given the results from both Kenya and Côte d'Ivoire, evidence indicates that even within the same trial, estimates of entomological efficacy varies. There was a relationship between the percentage reduction of mosquitoes caused by larviciding and the absolute reduction of mosquitoes, but even within the same studies or for matched species where entomological sampling and larval habitat is likely to be more consistent, the uncertainty in these associations were large (Fig 5F). Table C in S1 Text summarizes the estimated epidemiological impacts given the relative and absolute reductions in mosquito larval densities in each trail.

## Discussion

The theoretical model shows that LSM clearly has potential to complement vector control efforts and reduce malaria disease transmission across settings. Expectedly, the absolute reduction in adult mosquito densities achieved translates into different relative reductions in epidemiological outcomes depending on site-specific malaria endemicity, vector compositions and larval habitats, and residual transmission (in this modelling context, this is the degree that mosquitoes are assumed to blood feed outdoors), among other factors. The Kenya and Côte d-Ivoire trials represented two studies in

Africa with controlled before-and-after design that included ITNs in the control arms and reported both epidemiological and entomological metrics [34,35]. In a recent Cochrane review, another study from Tanzania had a similar format [44,45] but compared outcomes to untreated nets [36], but this is an unlikely situation moving forward as communities will be nearly universally using ITNs, so we chose not to include the study here. In this assessment, we were able to use the relative reduction in adult vector densities to parameterise a transmission model to broadly recreate empirical epidemiological outcomes at the village level or arm level. We discuss our limitations and challenges and how these findings could inform on the assessment frameworks for different forms of LSM.

Reductions in adult mosquito densities can be attributed to larvicide applications and deliver locally specific epidemiological outcomes. While it is key to note that all villages receiving larviciding benefited from the intervention, there was not a clear association between the estimated absolute number of mosquitoes reduced and the ultimate epidemiological outcome in the different villages in Kenya and Côte d'Ivoire. Although, there appears to be some linkage between mosquito density reductions and epidemiological outcomes (Fig 5), with higher burden settings showing generally higher absolute reductions in vectors using model derived estimates. There are several potential reasons for the differences that we observe. First, there were differences between villages in the use of ITNs, particularly in Kenya [34], that could complementarily contribute to transmission reductions. Similarly, activity of humans and vectors (rendering vector contact with ITNs less frequent or leaving people more exposed) that drive the degree of residual transmission will alter the potential impact from larvicide and ITNs [15]. Differences in the microclimate may exist that alter key elements such as the gonotrophic cycle [46,47], extrinsic incubation period [48] or other life traits of vectors [49]. The time from egg laying to pupation is variable with estimates having been reported to vary between 5 and 14 days [50], the trials applied larvicide either weekly [44] or fortnightly [35] so that if micro conditions differed between villages, which could have enabled different rates of passage through the aquatic life stages, adult mosquito emergence rates could also feasibly differ. In Côte d'Ivoire this may be a contributing explanation for the lower impact of larviciding efforts against adult densities in Nambatiourkaha, in contrast to Kakologo (Table 2).

The trial teams expected heterogeneity as many factors differed between sites that could not be standardized with restricted budgets. In Kenya, the valleys where villages were situated differed in their landscape; while the trial team were diligent in maximising consistent effort across villages, the variable habitats meant it was easier in some villages to achieve 100% coverage with larval habitat spraying and less so in others where permanent swamps maintained populations of *An. funestus*. Going forward, new technology may help mitigate for this as demonstrated recently in Rwanda [39]. The movement of vectors into villages was likely different across the villages depending on the surrounding terrain, and the current modelling exercise does not account for this. The size of the area treated versus the flight range of the mosquitoes can be a major contributor to the uncertainty around impact. Given the heterogeneity observed and the multiple reasons driving these differences, the use of control arms in non-randomized trials as comparison settings, or trials with low numbers of replicated villages, to quantify the potential impact from larviciding may be limited. Cluster-randomized trials would need to be very large to accommodate for the high variability in the effect size raising a question over their feasibility for environmental interventions and making high quality observational data critical [37]. Underlying factors could be driving local differences that are not isolated to the larvicidal impact. For the Kenyan villages, the transmission model systematically estimated higher prevalence burdens than empirically measured. The efficacy impacts from the model when comparing treatment village impact to the counterfactual - which holds all parameters constant except excluding larvicide effects – returned estimates of more than 50% reductions in prevalence across villages. This compares to a Cochrane review of larviciding that reported a 21% (11% -29%) reduction in parasite prevalence due to the treatment relative to the areas not receiving larviciding, after translating study results to relative risk ratios [51]. The discrepancy between model simulations and empirical observations was particularly evident in control settings (Fig 3). Large reductions in mosquito densities were empirically measured in control villages. It may be that the nets in this instance were more effective than predicted by the model parameters described. The mass net campaign in Kenya was coupled with educational programs

to encourage good practice for mosquito management in and around the home, and the measles vaccination program is likely to have reduced malaria burden in the local community too through similar educational messaging reiterating the need to actively reduce local mosquito breeding potential. Hydrological conditions were also different in the trial follow up period with dry spells and flash floods in the valley areas which made habitats likely less favorable for larval development and adult emergence. However, we did not try to capture interannual variation in rainfall and instead retained the same rainfall patterns – that drive vector density trends in the model – year-on-year. Taken together, these factors may explain the discrepancy in the transmission model outcomes and trial results, though other explanations are also possible. In all villages, the model – that can theoretically isolate impact comparing larvicide scenarios to matched counterfactuals – and empirical data comparing control and treatment villages, suggest impressive reductions in prevalence in children of 6-month to 10-years of age from the regular application of larvicide.

In Côte d'Ivoire, clinical incidence estimates across all age cohorts reported were far greater in the treatment arm than the control arm villages [35]. The mosquito counts were very similar however when considering either larvae or adults [35]. Given such contrasting incidence estimates at baseline, we would expect higher mosquito densities (following classical modelling theory [52,53]) in the locations with high malaria burden. Other settings – such as the Cascades region in Burkina Faso – have reported a similar paradigm where high mosquito densities and lower burden villages contrasted with lower mosquito density and higher burden villages [54]. The authors posited that an individuals' perceived risk of being bitten by infectious vectors differed when fewer mosquitoes were present, and that this may result in more transmission from fewer vectors [54]. Very high mosquito densities are found around rice irrigation and there is an established paradigm in these regions too where high vector counts do not translate to high burden [55,56]. This rice paradigm has been explained by higher incomes meaning better housing, access and use of ITNs, other prevention and treatment seeking in villages closer to the rice paddies [57,58]. The Côte d'Ivoire trial relied on retrospective health facility data with substantial village-level variability. In this modelling exercise, we tried to infer from these clinical incidence data on prevalence trends and impacts. There are well-documented challenges with using health facility data for trends analyses: treatment-seeking behaviour can differ between and among communities [59,60], there is uncertainty in the population sizes served by some health facilities which can complicate trend interpretation [7], there are different approaches to treating and counting patients presenting with fevers as malaria cases [60,61] and potential shortages or stock-outs in equipment that impact surveillance efforts [60–62]. These uncertainties represent a limitation for our analysis. As an alternative method antenatal care trends could be used [61] but were not available for our analysis.

The larval reductions were impressive in Côte d'Ivoire, using reductions determined from the larval data within the transmission model, greater impacts were estimated than the empirical data observed (Fig E and Table C in S1 Text). The trial noted that all larval sites that were findable were also treated. This indicates that any unfound larval habitat contributing to adult females in the region would not contribute to relative reduction in larval density estimates, thus potentially over-estimating impact. Another potentially critical metric is the proportion of larval habitats that are found that contain pupae (or late stage larvae). Not all larvae will successfully pupate to adult stage vectors, so even where larval densities are high, the emergence of adults may be hard to predict. A limitation of our theoretical modelling exercise is that we do not consider the variability in productivity between larval habitats, nor stages of larvae present. The exercise was used to show how the same effort in two places might achieve different relative reductions, however, it might be easier to find highly productive larval sites and managing these through LSM would then have greater benefit (would reduce the overall densities by a greater degree). Similarly, larval sites in a high transmission location may be generally more productive and rapidly found, so the absolute number of mosquitoes killed may be far higher than for low transmission locations per unit effort.

The theoretical modelling exercise demonstrates that across the endemicities tested (low, moderate and high transmission), the complementary use of larviciding and ITNs will likely have a greater relative benefit if residual transmission is high (Fig 1C, 1F, and 1I). The large impact observed in Côte d'Ivoire demonstrates this further, as the region has reportedly high

outdoor transmission [63]. The modelling exercise also demonstrates how large reductions in absolute numbers of vectors may result in small or undetectable relative reductions in mosquito densities with correspondingly small or undetectable impacts on epidemiological metrics. The high transmission situation in the flood plains of The Gambia [22] is sometimes reported as evidence against LSM in areas with large areas of larval habitat. The inaccessibility to reach some sites due to the terrain and hydrological events limited the percentage of reachable larval habitat. So, although the concerted effort to apply larvicides with ground teams in The Gambia had no measurable impact, the results should not deter efforts in other settings where the entomological and epidemiological situation might be far more amenable to LSM. This is particularly true as new technologies increase the probability of finding, reaching and treating larval habitat [39].

An important consideration is appropriately powered entomological sampling and the methods used to count mosquitoes. Sampling mosquitoes is notoriously challenging [64–67] and many studies are considered under-powered to detect changes in densities, sporozoite rates, foraging behaviours and other critical aspects of mosquito ecology that are associated with malaria transmission. Very clear reductions in adult densities, however, were measurable in the Kenyan and Côte d'Ivoire trials given the committed follow up effort from both trial teams. In Kenya, the trial authors noted that the use of pyrethrum spray catches for adult vectors may have underestimated absolute densities given the opportunity for mosquitoes to have left the house prior to catch teams operating in the morning [44]. This would be however a consistent concern across villages. Evidence suggests the method used to record adult densities and species compositions can return variable results, with sampling methods around housing tending to target anthropophilic species and sampling elsewhere perhaps better representing opportunistic species [68–70]. Therefore, the entomological method to employ to assess LSM impacts must consider these challenges, perhaps combining different approaches as was done in Côte d'Ivoire where by pyrethrum spray catches and window traps were used [35], though this particular method combination may still draw the focus onto house-dwelling species. Developing entomological methods that are highly powered and precise will prove key for monitoring environmental interventions. An alternative indicator that could be used is the proportion of habitats that contain late stage malaria vectors (pupae) at any given time, as a regular application, should prevent that and consequently lead to a reduction in adult vectors. Future work could consider how this metric might be predictive of epidemiological impacts.

The two trials and the theoretical modelling exercise all show that the health benefits of larviciding can be powerful. The model could be used to simulate impacts across malaria-endemic settings, but the outcomes from real efforts will be heavily led by the relative reduction in adult female mosquito densities ultimately achieved. How this positive theoretical result is achieved in reality is a key question and site-to-site variability in success would be highly dependent on effort, scale, LSM techniques adopted, and the various site characteristics discussed. Yet, in a resource-poor world, the question remains: what is the best way to assess LSM (including larviciding), to translate findings across different settings so that we can advise stake-holders and maximise control outcomes? The ecological and epidemiological data included in this assessment and for other studies [36,37,45], and the modelling results presented here and elsewhere [71,72] indicate that while variable, LSM can deliver impact. The where, when and how LSM, and any environmental intervention, will best deliver for communities is likely then dependent on good planning, local expertise of eco-epidemiological conditions, and well-financed task forces.

The Kenyan [34] and Côte d'Ivoire [35] examples indicate that for larviciding, measurable changes in the relative mosquito density before and after action translated to public health impact. In both trials, village level efforts covered at least 100 households demonstrating the necessity for concerted and sustained effort across a reasonable area for impact evaluation. Mosquito counts are a highly variable metric, and total numbers will be determined partly by weather conditions but also sampling effort, and the choice of method used for collections [73,74]. Year-on-year variability in conditions, or variable micro-climate and core intervention use among control and treatment villages could make it difficult to attribute impacts to particular actions, so data collection efforts could consider these elements for statistical assessment. A selection of well powered observational studies could generate invaluable information particularly for non-larviciding forms of LSM (those strategies that modify or manipulate habitat, or eliminate or remove larval habitats).

In Côte d'Ivoire, there was a larger reduction in larval densities (98%, CI 95–99%) than adults (81%, CI 67% – 90%), with adults better predicting the epidemiological impact. For interventions that do not need to find larval habitat [39], then surveys of randomly chosen larval sites could be sufficient to indicate true change. However, the tremendous variability in habitats within single locations would mean that, to be appropriately powered, we would need vast survey efforts so this is probably not feasible. We anticipate that the parsimonious modelling approach taken could reflect alternative mitigation strategies that presents a useful resource for future scenario investigation. However, a limitation is that it does not allow us to recreate explicit effects on aquatic life stages (we ignore density dependence) or detail stochastic elimination and resurgence events (such as migration of vectors into a setting from surrounding habitats) that likely play out spatially around settlements. We also do not simulate spillover effects or vector migration across untreated areas in the modelling exercise. Spatial modelling approaches [75] may be useful to consider how *Anopheles* dispersal, approach to aestivation, and recolonization following disturbance across communities could drive further heterogeneity in LSM impacts although little data currently exists on the transience of such ecological conditions. We would expect such challenges to affect LSM outcomes depending on how lasting implemented changes prove to be in real-world contexts, or how neighbouring locations are managing malaria vectors. Another limitation of the modelling is that we assume reducing mosquito densities does not alter the vector competence of the different species. It is still unclear how density dependent larval development impacts vector competence in field settings, though results from the laboratory suggest it could be substantial [76–78].

An ambitious pathway to sustained control has been laid out for malaria [79]. Sustainably managing habitat has benefit for climate, ecology and public health, particularly given the newly evidenced emissions costs of alternative interventions [80]. The work emphasizes the heterogeneity in the entomological impact of larviciding, as had previously been observed elsewhere [24,36], which can complicate the more widespread use of this technology. Nevertheless, the models demonstrates how well collected entomological data on adult mosquito abundance can be combined with transmission dynamics mathematical models to predict the epidemiological impact of LSM. One advantage of the modelling approach we have taken is that it is agnostic of the LSM strategy deployed as long as the impact achieves some reduction in mosquito densities. Further entomological, epidemiological and economic studies are needed in different settings to extend the evidence base to demonstrate how LSM can be best used as an effective malaria control tool.

## Methods

### Modelling to demonstrate theoretical LSM impact

As our baseline model, we use a model of malaria transmission dynamics [81–84] to demonstrate the potential public health impact of LSM. The *malariasimulation* model has been comprehensively described elsewhere [4,85] and the code is available (https://github.com/mrc-ide/malariasimulation) [86]. We outline the key assumptions needed to understand and reproduce the modelling framework.

The human component of the model is a non-spatial, stochastic, individual-based model of malaria transmission and the mosquito component is deterministic. In the human population, individuals are born susceptible to *Plasmodium falciparum* infection and are exposed to infectious mosquito bites at a rate that depends on local human and mosquito ecology and parasite prevalence in mosquitoes; maternal immunity is inherited by offspring, which subsequently decays in the initial 6 months of life. The risk of developing infection declines with age due to naturally acquired immunity following continual exposure. After exposure to infection, individuals are susceptible to clinical and severe disease and death [84,87].

In the transmission model, a key parameter is the number of mosquitoes relative to people that reflects different entomological inoculation rates (EIRs) and corresponding transmission endemicity levels. The equilibrium of this parameter is defined by a carrying capacity on larval mosquito stages.

**Mosquito model.** The vector model is based on the deterministic model previously described in White *et al.* [82]. Adult female mosquitoes emerge after progressing through larval and pupal stages [82]. The carrying capacity of the larval

population is allowed to vary according to local seasonality and determines the emerging adult mosquito density and, hence baseline transmission intensity in the absence of interventions.

Only female adult mosquitoes are explicitly modelled, and the adults that emerge are assumed susceptible to malaria infection. After emergence, the adults are assumed to mate and pursue a series of gonotrophic cycles, consisting of host-seeking (to blood-feed) and searching for aquatic habitats (in which to oviposit) and dying at an age-dependent rate in the absence of interventions. Mosquitoes can be simulated by parameterizing species-specific bionomics to broadly describe the behaviors that might be expected by different vectors. The proportion of bites taken on humans for different mosquito vectors are estimated following Massey et al. [88] and Killeen et al. [89]. The probability that bites are attempted indoors or in bed in the absence of indoor interventions is set at the value determined in Sherrard-Smith et al. [15]. Life expectancy, foraging and blood feeding rates are taken from Griffin et al. [81].

In this study, we extend the model to investigate the impact of interventions that reduce adult female mosquito densities. Our aim is to keep this impact general, so that some reduction in adult mosquitoes could be the result of a range of interventions. Interventions that mitigate the landscape are ecologically distinct to those that target the juvenile stages of the lifecycle, occurring in water bodies ("aquatic habitats") and we do not describe specific mechanisms for how these may differ, for instance the former may produce permanent changes in mosquito densities.

The details of how density dependent competition in juvenile populations regulates adult population sizes are currently not well understood, and many assumptions are based on artificial laboratory experiments. Despite this, there are potential associations between larval stage diet, habitat size, temperatures and other conditions such as stress induced food or water availability that alter the rate of larval development, and adult body size and speed of emergence which affect vector competence, fecundity and daily mortality [78,90–92]. Rather than make assumptions about the nature and form of these unknown density dependencies, we assume LSM interventions act by reducing the number of adult mosquitoes per person that can be sustained by the environment. The carrying capacity is impacted by the LSM effects, but all other (per capita) mosquito bionomics are set to be independent of LSM (i.e., mosquito adult life-expectancy remains constant for different levels of LSM).

**Insecticide treated net (ITN) interventions.** The degree of protection afforded by the presence of ITNs depends on the proportion of bites received when a person is in bed. ITNs are assumed to impact the probability that an *Anopheles* mosquito either successfully feeds, is repelled, or is killed on a feeding attempt. These probabilities have been determined for different ITN classes from systematic reviews of phase II experimental hut trials [41,42,93]. These probabilities change as the mortality inducing effects from the insecticidal active compound wanes [81].

ITN usage can be set for any simulation and falls exponentially from the moment of distribution. Recent work indicates that net use wanes relatively rapidly for countries in sub-Saharan Africa [9]. In our simulations, ITNs are simulated so that after 5 years, half as many people use nets as when immediately after the initial distribution.

**Larval source management.** As discussed, in *malariasimulation,* the vector model initialises vector species with a given adult female mosquito density that is determined by the user-defined EIR. This caps the population size of mosquitoes that can be supported within the environment and ultimately determines the transmission intensity for each simulation. This baseline total number of mosquitoes of any defined species in the system can change over time. The model includes seasonality so that rainfall cycles can be matched to drive intra-annual variation. We introduced additional functionality to model a change in the absolute or relative number of mosquitoes of a specified species at a point in time to reflect population suppression due to LSM activity. This immediately elicits a reduction in the number of mosquitoes from its original equilibrium value to the newly selected value.

**Seasonality.** To incorporate seasonality, a time-varying carrying capacity is used:

$$K(t) = K_0 \frac{R(t)}{R}$$

(1)

where $K_0$ is the carrying capacity, $\overline{R}$ the mean annual rainfall and $R(t)$ is a seasonal curve, which is a function of time. A Fourier transform function, representing the first 3 frequencies, was previously fitted to rainfall data from the US Climate Prediction Center [94,95] for sub-Saharan Africa between 2002 and 2009, and this approach was used to estimate:

$$R(t) = g_0 + \sum_{i=1}^{3} g_i \cos(2\pi ti) + h_i \sin(2\pi ti)$$

(2)

Eq. 2 results in seasonal variation in the mosquito birth rate, altering adult mosquito densities through the transmission season to parallel the expected rainfall patterns in a location.

### Theoretical potential of LSM

In this parameter sensitivity exercise, the general epidemiological impact of LSM is explored using the *malariasimulation* model. Theoretical seasonal settings are simulated with 60% of the population using ITNs immediately after the mass campaign. The potential of LSM given different residual transmission profiles can be explored by selecting vectors with reduced propensity to blood-forage on people indoors or in bed. Here, simulations are shown for a population with low (c.8% *Plasmodium falciparum* positivity detected using microscopy in children of 6–59 months of age), medium (c.30%) or high (c.50%) transmission measured as microscopically detectable *P. falciparum* prevalence in children of 6 – 59 months of age across one year preceding the introduction of LSM. The effect of pyrethroid resistance can be explored using parameters that reduce the initial and enduring efficacy of ITNs to suppress bites and kill blood-foraging mosquitoes [42]. Scenarios are investigated with different pyrethroid resistance profiles for simulated mosquito populations ranging from 0% (all mosquitoes are killed on exposure to a discriminating dose of pyrethroid in the bioassay test) to 90% (where all mosquitoes survive exposure to a discriminating dose in the bioassay test). These profiles translate into different potential impacts from pyrethroid-ITNs ([42], updated in [41,96]). In parallel, the potential of LSM to act when residual transmission is higher, rendering the potential of indoor interventions less powerful, is investigated for a range of outdoor biting vector profiles where mosquitoes are simulated so that 40% to 99% of bites are attempted when a person is in bed. LSM strategies are introduced that reduce the numbers of all female *Anopheles* by 60% in the low transmission scenario. Arbitrarily, this corresponds to a model simulation that has a baseline 15.4 mosquitoes per person in the simulation prior to LSM actions and 6.2 mosquitoes per person afterward. The absolute number of mosquitoes that this corresponds to is calculated and used to investigate the same absolute reduction in mosquito numbers but in the moderate and high transmission settings. The potential impacts are contrasted to simulations showing a relative reduction of 60% in these higher transmission settings. In each case, no waning impact for the LSM action is implemented. The relative efficacy against malaria burden – here using prevalence in children of 6 – 59 months of age – of mass ITNs or mass ITNs and sustained LSM across the years 0–3 (from a mass ITN campaign onward), are compared to a scenario where ITNs are no longer deployed at year 0. In the counterfactual simulation, malaria burden generally resurges.

### Simulating empirical trials for larviciding

Trials in Kenya [34] and Côte d'Ivoire [35] attempted to measure entomological and epidemiological changes brought about by using *Bti*-larviciding. Next, we try to simulate these efforts using the transmission model and explore some of the associated challenges.

### Data resources

**Kenya [34].** The trial was conducted in 2005 in the Kakamega and Vihiga Districts of Kenya to test whether microbial larviciding with a *Bacillus*-based larvicide could achieve reductions in malaria burden [34]. Study sites are described

previously [34]. Briefly, six highland valley communities were recruited for the study. The region has both a long and a short rainy season and is characterized by small streams and papyrus swamps within the valleys of the hilly region. Microbial larviciding was applied every week to all identified water bodies in 3 of the 6 communities from July 2005 for 19 months. Monthly collections of indoor resting mosquitoes were made using pyrethrum spray from 10 randomly selected sentinel houses in each village and vectors from the *An. gambiae* s.l. complex were individually identified to species using polymerase chain reaction [34]. Alongside the adult entomological data, a cohort of children aged from 6-months to 10-years were tracked throughout the trial to assess the public health impact of the larvicidal intervention, rapid malaria tests (OptiMal DiaMed, Cressier, Switzerland) determined positivity, and microscopic parasite identification determined plasmodial species and infection density [34].

The study design and the timing of intervention pressures in each of the six sentinel villages in the study are documented [34]. The data were analysed to estimate the absolute and proportional reduction in adult mosquito densities observed before and after the intervention of either ITNs or both ITNs and larviciding. Prior to the larvicidal intervention, there were very minimal vector control interventions present in the region. A mass campaign with pyrethroid ITNs was delivered at the same time as the start of the larvicidal intervention and net coverage was further raised the following year as ITNs were distributed to all children together with a measles vaccination campaign [34]. The reductions in prevalence observed were then possibly driven by both the increased coverage of ITNs in all villages (approximately 50% net use post June 2005 reported [34]), and due to larviciding in the three sentinel villages receiving the weekly application of *Bti*-larvicide. To try and ensure no spillover impacts from larviciding, the villages were chosen within valleys that were approximately 2–4 km² in area and at least 1 km apart [34].

**Côte d'Ivoire [35].** The trial in Côte d'Ivoire considered 4 villages of at least 100 households in size, with an average of 6.1 people per household, in the department of Napiélédougou in Korhogo Health District where communities had received pyrethroid-ITNs via the mass campaign that took place in 2017 [35]. The trial team were reactive to local weather conditions, conducting larviciding outside the times of heavy rain and applying effort given perceived need, applying 22 larviciding treatment events across the year [35]. Rice paddies and reflows from rain showers held about half of all breeding sites, but earthenware vessels, river edges, pools and puddles, village pumps, hoof prints, swamps, jars, ponds and watering wells also contributed [35]. Two villages (Kolékaha and Lofinékaha) were monitored as comparator settings and assumed to have similar burden to the two villages (Kakologo and Nambatiourkaha) selected to receive *Bti*-larviciding treatments fortnightly for the following year; March 2019 to February 2020 [35]. Buffer spaces of 2 km in radius around the treatment villages were also treated [35].

Mosquito monitoring every two months throughout the trial showed reductions of all aquatic stages in the treated villages relative to comparators and compared to baseline [35]. Adult *Anopheles* densities were monitored from 10 randomly selected households over two consecutive nights using window traps and pyrethrum spraying, and these surveys were repeated in each village every 2 months. *Anopheles gambiae* s.l. dominated (75%) of which most were *gambiae s.s.* and 4% were *coluzzii,* and represented all vectors at baseline, but later *An. funestus, An. nili* and *An. pharoensis* were reportedly present [35]. Of 357 mosquitoes sampled that had blood fed (the origin of the blood meal was not known in 32 specimens – explained further in [35], of 389 screened), 333 had human blood meals (313 solely on humans, 20 with both human and livestock blood meals identified) [35]. This estimate aligned well with previous reviews specifying human blood index estimates for *An. gambiae* and *An. funestus* of 0.92 and 0.94 respectively [88,89].

## Statistical analysis

To estimate change in mosquito density before and after interventions a negative binomial regression was fitted to the female Anopheles density data with a binary covariate signalling whether counts were scored pre- or post- intervention (covariate $x_1$), a term signalling treatment group (larviciding or non-larviciding arm, covariate $x_2$), and in the case of the Kenya adult mosquito data, a covariate for species (*An. funestus* or *An. gambiae, $x_3$*) and the full complement of their

interactions. The difference-in-differences approach was taken to estimate the additional benefit that can be attributed explicitly to larviciding. The first model (Eqs 3 and 4) holds deviations at the village level constant in matrix **Y,** and also encodes deviations in the predictors across the number of people per household for the Côte d'Ivoire trial data, so we get:

$$\theta_i = e^{\beta_0 + \beta \mathbf{X} + b \mathbf{Y}} \tag{3}$$

$$Y_i \sim NB(\mu_i \theta_i, \varphi) \tag{4}$$

The matrix **X** can be expanded to:

$$log(\theta_i) = \beta_0 + \beta_1 x_{1i} + \beta_2 x_{2i} + \beta_3 x_{3i} + \beta_4 x_{1i} x_{2i} + \beta_5 x_{1i} x_{3i} + \beta_6 x_{2i} x_{3i} + \beta_7 x_{1i} x_{2i} x_{3i} + (1|village) \tag{5}$$

The coefficient $\beta_0$ corresponds to the intercept. The coefficients $\beta_{1-3}$ reflect the difference of post intervention (where $x_1$ takes the value 1) to pre intervention ($x_1 = 0$), LSM treatment ($x_2 = 1$) to control arm ($x_2 = 0$), and *An. gambiae* to *An. funestus* (reference group) respectively. The coefficients $\beta_{4-7}$ then indicate the interactions between these terms. The village level deviations are noted by intercept $\beta_0$. The model of the Côte d'Ivoire larval and adult data simplifies given no species specific information for larval counts is known and the species composition of adults was dominated by *An. gambiae* (94.7%, n = 2732). This statistical model (Eqs 3–5) is fitted with a Bayesian framework using R and the package rstanarm [97] using the *stan_glmer.nb* function [98]. Convergence of the MCMC was diagnosed visually checking chains and checking that $\widehat{R} \leq 1$.

To estimate the difference-in-differences $\partial_{DiD}$, it is necessary to translate the model outputs to predict the proportional reduction in mosquitoes while accounting for the change in mosquito numbers pre- and post-intervention within the controls relative to treatments [99]. For *An. funestus,* we get:

$$\partial_{DiD,fun} = 100 \times (1 - \exp^{[(\beta_0 + \beta_2) - (\beta_0 + \beta_1 + \beta_2 + \beta_5)] - [(\beta_0) - (\beta_0 + \beta_1)]}) \tag{6}$$

Similarly, for *An. gambiae*:

$$\partial_{DiD,gam} = 100 \times (1 - \exp^{[(\beta_0 + \beta_1 + \beta_2 + \beta_3 + \beta_4 + \beta_5 + \beta_6 + \beta_7) - (\beta_0 + \beta_2 + \beta_3 + \beta_6)] - [(\beta_0 + \beta_1 + \beta_3 + \beta_5) - (\beta_0 + \beta_3)]}) \tag{7}$$

These simplify for the Côte d'Ivoire data given no species covariate is included.

A second model is fitted to determine the village specific impacts relative to the control arm data. In this case, we recoded the covariate for arm ($x_2$) to a logical variable of 0 (control) or 1 (LSM). The model takes the form:

$$\theta_i = e^{\beta_0 + \beta \mathbf{X}} \tag{8}$$

$$Y_i \sim NB(\mu_i \theta_i, \varphi) \tag{9}$$

Where matrix **X,** matches Eq 5, but replaces the arm covariate with the additional details for the LSM-village level differences. The second statistical model (Eqs 8 and 9) is also fitted with a Bayesian framework using R and the package rstanarm [97], this time applying the *stan_glm.nb* function [98]. Posterior checks for statistical models are provided in Table A in S1 Text.

In each case, uncertainty on $\partial_{DiD,species}$ was generated by randomly sampling the posterior predictive draws 1,000 times and generating 90% credible intervals. These statistical models estimate an explicit impact (proportional reduction in mosquito counts) attributable to larviciding that are then used for the transmission model parameterization (Table 1).

## Transmission model simulations of trial data

By calibrating the transmission model to the pre-trial baseline prevalence, we have a starting point from which to reduce the adult female mosquito densities. We simulate *An. gambiae*-like mosquitoes and *An. funestus*-like mosquitoes for the trial using systematic reviews to parameterize the bionomics of each species.

**Kenya trial.** Prevalence in children of 6-months to 10-years of age was recorded intermittently across the Kenya trial (S1 Data) before and after the interventions began in July 2005. To include the effects of larviciding on reducing mosquito densities, the difference-in-difference analysis was applied separately for *An. gambiae* s.l. and *An. funestus* s.l. for each larviciding village as compared to all non-larviciding villages as noted above (Eq 3).

The adult mosquito density per human is, in general, an unknown, yet crucial quantity for determining transmission intensity. As such, this parameter was chosen so that modelled average prevalence in children of 6-months to 10-years of age across the 3 sites for each arm matched the observed quantity as measured, on average, between January 2004 and June 2005, before the *Bti* larvicide was introduced in July 2005.

For the Kenyan trial, we simulate mosquito net use as it progressively increased in each village from the onset of the trial observational period (February 2004 – January 2007) (Fig B in S1 Text). We assume that adherence to using nets is relatively high so that it takes 5 years before half as many people are using the net relative to those using ITNs immediately after the initial distribution, with net use dropping exponentially at this rate. This gives the usage patterns shown in Fig B in S1 Text for each village respectively. Some households in the Kenyan trial reported privately spraying insecticide or using coils to deter mosquitoes locally but we did not simulate these in the modelling exercise.

Previous drug treatment for the human population is unknown for the study region but the cohort of children tracked during the CRT [46] received Sulfadoxine-pyrimethamine between April and June 2004 and subsequently artemether-lumefantrin (AL) combination therapy. We assumed equivalent treatment was in operation across all villages such that until June 2004, we simulated 5% of the population were receiving AL and 35% were receiving SP treatment for clinical cases. From July 2005, we simulated that 40% of people received AL with 10% receiving SP across the whole population.

To generate uncertainty in the simulations, a sensitivity analysis is included. Intervention impacts are included as the ITN mortality inducing effect at the deployment of a the mosquito net ($dn_0 = 0.34$, range 0.28-0.41 following previous work in [40]); the mean duration of action of the ITNs given an exponential decay in induced mortality ($gamma_n = 2.64$, $2.0 - 3.0$ years, [40]); the probability of mosquitoes attempting to feed when a person is in bed in the absence of interventions (for *An. gambiae*: 0.85, $0.80 - 0.92$; for *An. funestus*: 0.78, $0.73 - 0.83$, using parameter ranges from [15]); and the preference to feed on humans for a blood meal (for *An. gambiae*: 0.92, $0.85 - 0.98$; for *An. funestus*: 0.94, $0.84 - 0.98$, using parameter ranges from [89]).

The predictive performance of the approach is evaluated by plotting the linear model estimated density of mosquitoes against the trial data and then comparing model predicted prevalence in children 6-months to 10-years of age to trial observations. At each time point where prevalence was measured (Nov 2005–Jan 2006, May 2006–July 2006, and Nov 2006-Jan 2007), we can compare modelled versus observed efficacy against prevalence:

$$E_t^x = \frac{\left(P_{C,t}^x - P_{T,t}^x\right)}{P_{C,t}^x},$$

(10)

where $P_C$ is the prevalence estimate for the control arm, $P_T$ is prevalence of the larvicidal arm and $E$ is the efficacy as determined by either the observed data ($x = 1$) or model data ($x = 2$) at time $t$.

**Côte d'Ivoire trial.** In the Côte d'Ivoire trial, the ITN coverage immediately after the campaign was reported to be over 80% [35] which we assume to have reduced exponentially to about 40% cover two years later when the larviciding began, falling to 25% at the end of the follow up year. This aligns with the trial reported ITN use of 40.7% in the ITN and larviciding villages, and 36.2% in the ITN only villages averaged over the follow up year [35]. To explore the potential

impact from ITNs we conducted a sensitivity analysis for the range of phenotypic pyrethroid resistance scenarios from 60% to 95% in the absence of bioassay data from the site and in the knowledge that recent analysis of mosquito vectors in the region showed high prevalence of genetic markers for pyrethroid resistance in the vectors *An. gambiae* s.s. and *An. coluzzi*, and seasonal abundance patterns aligning with the rains given the dominance of *An. gambiae* s.l. [63]. Community activities were reported that left people susceptible to outdoor transmission [63], therefore we simulated a range of estimates for the proportion of bites taken indoors (0.75 to 0.9) and in bed (0.7 to 0.85).

The transmission model is parameterized for each of the 4 villages in Côte d'Ivoire [35] that applied ITNs with or without microbial larvicide to reduce mosquito densities. The trial did not record an explicit time point estimate of prevalence or clinical incidence, but did record the sum total of cases and population numbers for each village. The model assumes a seasonal transmission in mosquitoes that corresponds to burden. To calibrate the model to a representative level of burden for the 4 villages, we created a trend using this seasonal profile across space and time and weighted this by the numbers of people in each village. This returned estimated relative all-age prevalence levels, as estimated by light microscopy, in August 2017 of 34%, 24%, 42% and 27% in Kolékaha and Lofinékaha (control villages), and Kakologo and Nambatiourkaha (treatment villages) respectively. This aligned reasonably with the estimated aggregated cases in the baseline year for each village (278, 215, 426, and 339 total cases recorded at Health Facilities for the same villages respectively) given the estimated variability in population sizes (496, 553, 628 and 758 people counted in the 100 households per village, data courtesy of Tia *et al.*) [35].

At the trial arm level, clinical incidence for all-ages was 188.0 cases per 1,000 people per year between March 2018 and February 2019 for control villages, and 186.1 cases per 1,000 people per year in the March 2019 to February 2020 follow up period [35]. In the treatment arm, there were 291.8 cases per 1,000 people per year at baseline, this dropped to 111.4 cases per 1,000 people per year in the follow up period [35].

The model was used to output estimates for light microscopy prevalence in all ages, and all-age clinical incidence. Simulated estimated outcomes at the village level were compared visually to derived estimates of prevalence, and reported annual clinical incidence from trial information (Fig 5).

## Supporting information

**S1 Data. Xlsx file including data for Fillinger et al. 2009 [34].**
(XLSX)

**S1 Text. Table A.** Summary of the statistical models fitted to the Kenyan entomological data. **Table B.** Summary of the statistical models fitted to the Côte d'Ivoire entomological data. **Table C**. Summary outputs from transmission model simulations using reductions in larval mosquito density as estimated empirically at the village level in Côte d'Ivoire, uncertainty in parentheses show 90% uncertainty interval. **Fig A.** Theoretical impact of larval source strategies that suppress *Anopheles* mosquitoes. **Fig B.** The observed data for six sentinel villages tracked during a larviciding randomized control trial in Kenya [34]. **Fig C.** Net use measured in the tracked cohort of children of 6-months to 10-years of age at cross sectional surveys throughout the Kenyan trial. **Fig D.** Summary results for the crude difference-in-difference estimates. **Fig E.** Larval densities analysis and model simulated results for the Côte d'Ivoire trial.
(DOCX)

## Author contributions

**Conceptualization:** Ellie Sherrard-Smith, Ben Lambert, Thomas S. Churcher.

**Data curation:** Ellie Sherrard-Smith, Ulrike Fillinger, Jean-Philippe B. Tia, Benjamin G. Koudou, Emile S. F. Tchicaya.

**Formal analysis:** Ellie Sherrard-Smith.

**Funding acquisition:** Ellie Sherrard-Smith.

**Investigation:** Ellie Sherrard-Smith, Peter Winskill, Antoine Sanou, Fredros Okumu, Mercy Opiyo, Arran Hamlet, Ben Lambert.

**Methodology:** Ellie Sherrard-Smith, Peter Winskill, Giovanni Charles, Ben Lambert.

**Project administration:** Ellie Sherrard-Smith.

**Resources:** Ulrike Fillinger, Jean-Philippe B. Tia, Peter Winskill, Benjamin G. Koudou, Emile S. F. Tchicaya, Arran Hamlet, Giovanni Charles.

**Software:** Giovanni Charles.

**Supervision:** Ulrike Fillinger, Peter Winskill, Benjamin G. Koudou, Antoine Sanou, Fredros Okumu, Mercy Opiyo, Silas Majambere, Arran Hamlet, Ben Lambert, Thomas S. Churcher.

**Validation:** Ulrike Fillinger, Jean-Philippe B. Tia.

**Visualization:** Ellie Sherrard-Smith, Ulrike Fillinger, Jean-Philippe B. Tia.

**Writing – original draft:** Ellie Sherrard-Smith.

**Writing – review & editing:** Ellie Sherrard-Smith, Ulrike Fillinger, Jean-Philippe B. Tia, Peter Winskill, Benjamin G. Koudou, Emile S. F. Tchicaya, Antoine Sanou, Fredros Okumu, Mercy Opiyo, Silas Majambere, Arran Hamlet, Giovanni Charles, Ben Lambert, Thomas S. Churcher.

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
