## [Decision Letter · Decision Letter 0]

PPATHOGENS-D-25-00474

Heterogeneous impacts from larviciding across villages and considerations for monitoring and evaluation

PLOS Pathogens

Dear Dr. Sherrard-Smith,

Thank you for submitting your manuscript to PLOS Pathogens. After careful consideration, we feel that it has merit but does not fully meet PLOS Pathogens's publication criteria as it currently stands. Therefore, we invite you to submit a revised version of the manuscript that addresses the points raised during the review process.

Please submit your revised manuscript within 30 days Jun 18 2025 11:59PM. If you will need more time than this to complete your revisions, please reply to this message or contact the journal office at plospathogens@plos.org. Please include the following items when submitting your revised manuscript:

We look forward to receiving your revised manuscript.

Kind regards,

Kenneth Vernick

Academic Editor

PLOS Pathogens

Jeffrey Dvorin

Section Editor

PLOS Pathogens

Sumita Bhaduri-McIntosh

Editor-in-Chief

PLOS Pathogens

orcid.org/0000-0003-2946-9497

Michael Malim

Editor-in-Chief

PLOS Pathogens

orcid.org/0000-0002-7699-2064

**Journal Requirements:**

1) Please provide an Author Summary. This should appear in your manuscript between the Abstract (if applicable) and the Introduction, and should be 150-200 words long. The aim should be to make your findings accessible to a wide audience that includes both scientists and non-scientists. Sample summaries can be found on our website under Submission Guidelines:

https://journals.plos.org/plospathogens/s/submission-guidelines#loc-parts-of-a-submission

- ® on page: 33 and 44.

5) Please ensure that the funders and grant numbers match between the Financial Disclosure field and the Funding Information tab in your submission form. Note that the funders must be provided in the same order in both places as well.

**Reviewers' Comments:**

Reviewer's Responses to Questions

**Part I - Summary**

Reviewer #1: Very interesting and well-written manuscript on the impact of larviciding (as a supplementary tool, on top of bednets) on both mosquito abundance and malaria prevalence. It is unfortunate that the model did not (always) capture the observed trial data from Kenya and Cote d’Ivoire, but the authors rightly state that all villages receiving larviciding benefited from the intervention, and have done a wonderful job describing why modeling the impact of LSM is challenging. Their suggestions on how to improve entomological surveillance and measure other indicators are helpful.

I only have a few questions/comments:

Reviewer #2: The manuscript is providing indirect method or model for theoretical epidemiological benefits/impacts related to introduction of larval source management to supplement the core vector control intervention, the ITNs for malaria control and achieving its elimination. The study is cost effective as it can rely only on entomological parameters such as density of adult malaria vectors to predict the impact on averted malaria cases.

Reviewer #3: The authors present a compelling analysis of larval source management (LSM) as a complementary intervention for malaria control, using an individual-based transmission model parameterized with empirical data from two well-characterized trials in Western Kenya and Côte d’Ivoire. The study is methodologically strong, timely, and relevant for both implementation science and vector control policy.

The manuscript makes important contributions by showing that empirically measured reductions in adult mosquito densities—despite site-level heterogeneity—can be used to parameterize dynamic transmission models and broadly reproduce observed epidemiological outcomes. This is particularly valuable in a field where direct epidemiological evidence for LSM remains limited.

The manuscript is generally well-written, thorough, and scientifically sound. That said, I believe the following revisions and clarifications would strengthen the work and increase its value to the broader research community.

**Part II – Major Issues: Key Experiments Required for Acceptance**

Reviewer #1: It is not clear to me why only one LSM strategy (that reduces the number of female Anopheles by 60%) is used. What is your rationale? What would the outcome be if its impact is higher or lower? I also do not understand why then a reduction in the absolute number of mosquitoes by the same number as a 60% reduction in the low transmission setting is used. I could not find the reasoning for that.

I would ask the authors to copy some critical information (which is now in the methods section) to the results section, so the readers understand the context early on. For example, can you add the type of bednet, period(s) and frequency of larviciding (weekly Kenya, every fortnight Cote d’Ivoire) before showing the relative reduction in adult mosquito densities?

Discussion: You may want to add the (likely) increased relative impact of proper LSM strategies (on top of bednets) in areas with resistant mosquitoes and/or areas where mosquito behaviors reduce contact with ITNs/IRS.

Discussion: Throughout the text there is a wealth of information on confounding factors and suggestions on how to improve surveillance. I would ask the authors to create a box, summarizing the critical indicators that should (ideally) be measured to improve modeling exercises, and how those may differ between low/medium/high transmission settings.

Reviewer #2: I have only minor issues which will be addressed by the authors of the manuscript.

Reviewer #3: Major Comments

1. Clarify the rationale for using adult mosquito density over larval metricsWhile the discussion acknowledges practical limitations of using larval data, the manuscript could better articulate why adult densities—rather than larval or pupal indices—were chosen as the principal metric for model parameterization. Given that larval productivity varies and not all larvae translate into adults, readers would benefit from a clearer explanation of the trade-offs involved in this decision, especially for settings with complex aquatic habitats.

2. Expand discussion on model validation in Côte d’Ivoire.The Côte d’Ivoire trial relied on retrospective health facility data with substantial village-level variability. While the authors attempt to calibrate model outputs to estimated prevalence, further discussion on the robustness of these estimates—especially considering health-seeking behavior, facility reporting rates, and catchment uncertainties—would be valuable. Are alternative validation strategies or uncertainty propagation approaches feasible?

3. Strengthen the link between entomological reduction and transmission heterogeneity.The model performs well in explaining broad trends, but the linkage between mosquito density reductions and heterogeneous epidemiological outcomes could be expanded. For instance, do variations in species composition, net use, or behavioral plasticity in biting patterns explain the differing impacts across villages? Figure 5F suggests some signal, but the interpretation could be more clearly developed in the discussion.

4. Discuss spatial dynamics and vector movementThe assumption of spatial isolation is acknowledged, but the model does not simulate spillover effects or vector migration across untreated areas. While this is understandable for tractability, a short discussion on the implications of this omission—particularly for scalability or real-world program design—would add nuance.

5. Comment on implications for non-larviciding forms of LSMThe manuscript emphasizes larviciding, but the introduction and discussion frame LSM more broadly (including habitat manipulation and removal). A short comment on whether these findings are likely to translate to non-chemical approaches would increase the manuscript's applicability to integrated vector management (IVM).

**Part III – Minor Issues: Editorial and Data Presentation Modifications**

Reviewer #1: L254 (figure 1)

Fig 3H: y-axis label partially overlaps with values

L480 ‘unlike situation moving forward’ can you explain what you mean?

623-625 plus the residual efficacy of the product (?)

Reviewer #2: I would appreciate the contribution of the authors by elucidating the added theoretical epidemiological benefit of larval source management to the ITNs, the primary core vector control intervention for malaria control by using indirect method, an individual-based transmission model for falciparum malaria. They used peered reviewed information from published data sets from Kenya and Cote d’Ivoire and presented the outcomes as well as the limitations that will guides future researchers.

Therefore, here below are minors concerns that require corrections or clarifications in order to improve the manuscript:

Abstract

In line 56, In Kenya, adult mosquitoes were sampled using Pyrethrum Spraying Catching (PSC) method and not CDC-light traps. Similarly, to mention the methods used for sampling adult mosquitoes in Cote d’Ivoire: Standard Window Traps and Pyrethrum Spraying Catching.

Introduction

The introduction is well written and documented, but some below minor edits have to be considered.

- Line 76: To review the sentence as “the mass drug treatments is not a key intervention for malaria control. The referenced paper mentioned: " Prompt treatment of clinical malaria cases”.

- The last sentence between lines 79-81, requires a reference (see review copy highlighted in orange”.

- The last sentence between 143-147 looks long and has to be spilt into two sentences. “first sentence ending by ----- conditions, and the second starting from ---- It is also hard….”

- Line 155: At the sentence starting from line 155, the author used the word “suppression” of adult female mosquito density… The same word was used in other sections of the manuscript. This word seems not appropriate for malaria control strategy as most of core interventions aim the reduction of vector and malaria infection parameters. To find out an alternative word to replace “suppression”.

- Other edits are proposed on respective lines 163 and 168.

Results: The results are supported by Figures and Tables. The following are the minor clarifications or changes:

- The types of ITNs distributed and evaluated in each country have to be included in the narrative text of results section (Standards PermaNet2 in Cote d’Ivoire; PermaNet and Standard Olyset Net in Kenya).

- The annotations of figures (1,2,3,4,5) and tables (1,2). are mixing some narrative sentences which may be relocated in results or material&method sections. That means to shorten the annotations and keep only the naming of title, panels and other key figures of table and figures.

- Figures 2 and 4 are missing annotations of cited panels.

- The legend of Panel E, Figure 1 is not readable and needs the increase of the front size.

- The tables 1 and 2 require the improvement of formatting and improvement of presentations.

- Other minor corrections are proposed at the lines 334, 386.

Discussion

This section is well written and documented. Some corrections/clarifications proposed at the lines 608 and 617.

Material & Methods and statistical analysis.

Some corrections are highlighted and proposed at the following lines: 674, 792, 804/805, 973 and 978.

Reviewer #3: Minor Suggestions

* Figure 5: Panels are rich in information but could benefit from clearer legends or brief panel-specific descriptions in the caption, particularly for less familiar readers.

* Data accessibility: While code is well documented, the manuscript would benefit from a clearer summary in the main text of which datasets are fully publicly available and which are available upon request.

PLOS authors have the option to publish the peer review history of their article (what does this mean? ). If published, this will include your full peer review and any attached files.

**Do you want your identity to be public for this peer review?** For information about this choice, including consent withdrawal, please see our Privacy Policy .

Reviewer #1: No

Reviewer #2: No

Reviewer #3: No

**Figure resubmission:**
---

## [Editor Report · Decision Letter 1]

Dear Dr. Sherrard-Smith,

We are pleased to inform you that your manuscript 'Heterogeneous impacts for malaria control from larviciding across villages and considerations for monitoring and evaluation' has been provisionally accepted for publication in PLOS Pathogens.

Best regards,

Kenneth Vernick

Academic Editor

PLOS Pathogens

Jeffrey Dvorin

Section Editor

PLOS Pathogens

Sumita Bhaduri-McIntosh

Editor-in-Chief

PLOS Pathogens

orcid.org/0000-0003-2946-9497

Michael Malim

Editor-in-Chief

PLOS Pathogens

orcid.org/0000-0002-7699-2064

Referring to the below quoted exchange in the response letter, we suggest that the simple and frequently employed way to provide a synthetic summary would be as a final main text Figure or Supplementary Figure. This would not disrupt the text call-outs for the previous figures. We will leave it to the authors' discretion whether they think a summary figure would add value to the article. If you should decide to create a synthetic figure, the editors would need to approve it, but it would not require another review of the manuscript. If you decide not to create another figure, you can proceed as noted to final production of the accepted article.

"Discussion: Throughout the text there is a wealth of information on confounding factors and

suggestions on how to improve surveillance. I would ask the authors to create a box,

summarizing the critical indicators that should (ideally) be measured to improve modeling

exercises, and how those may differ between low/medium/high transmission settings.

We considered the Box 1 suggestion, however we could not find any guidance for including

this within the journal guidelines nor an online example of other manuscripts from this

publication including such a Box. We are open to the suggestion but request editorial

guidance on whether it is acceptable. Thank you for the suggestion and we hope that at

least we have included critical points within the discussion."